# Synergistic roles of Synaptotagmin-1 and complexin in calcium-regulated neuronal exocytosis

Sathish Ramakrishnan[1], Manindra Bera[1], Jeff Coleman[1], James E Rothman[1]*, Shyam S Krishnakumar[1,2]*

[1]Department of Cell Biology, Yale University School of Medicine, New Haven, United States; [2]Department of Clinical and Experimental Epilepsy, Institute of Neurology, University College London, London, United Kingdom

**Abstract** Calcium ($Ca^{2+}$)-evoked release of neurotransmitters from synaptic vesicles requires mechanisms both to prevent un-initiated fusion of vesicles (clamping) and to trigger fusion following $Ca^{2+}$-influx. The principal components involved in these processes are the vesicular fusion machinery (SNARE proteins) and the regulatory proteins, Synaptotagmin-1 and Complexin. Here, we use a reconstituted single-vesicle fusion assay under physiologically-relevant conditions to delineate a novel mechanism by which Synaptotagmin-1 and Complexin act synergistically to establish $Ca^{2+}$-regulated fusion. We find that under each vesicle, Synaptotagmin-1 oligomers bind and clamp a limited number of 'central' SNARE complexes via the primary interface and introduce a kinetic delay in vesicle fusion mediated by the excess of free SNAREpins. This in turn enables Complexin to arrest the remaining free 'peripheral' SNAREpins to produce a stably clamped vesicle. Activation of the central SNAREpins associated with Synaptotagmin-1 by $Ca^{2+}$ is sufficient to trigger rapid (<100 msec) and synchronous fusion of the docked vesicles.

*For correspondence:
james.rothman@yale.edu (JER);
shyam.krishnakumar@yale.edu
(SSK)

**Competing interests:** The authors declare that no competing interests exist.

## Introduction

The controlled yet rapid (sub-millisecond) release of neurotransmitters stored in synaptic vesicles (SVs) is central to all information processing in the brain (*Südhof, 2013*; *Kaeser and Regehr, 2014*; *Rizo, 2018*). Synaptic release of neurotransmitters relies on the efficient coupling of SV fusion to the triggering signal – action potential (AP)-evoked $Ca^{2+}$ influx into the pre-synaptic terminal (*Kaeser and Regehr, 2014*; *Südhof, 2013*). SV fusion is catalyzed by synaptic SNARE (soluble N-ethylmaleimide-sensitive factor attachment protein receptor) proteins, VAMP2 on the vesicles (v-SNAREs) and Syntaxin/SNAP25 (t-SNAREs) on the pre-synaptic membrane (*Söllner et al., 1993*; *Weber et al., 1998*). On their own, SNARE proteins are constitutively active and intrinsically trigger fusion in the range of 0.5–1 s (*Ramakrishnan et al., 2018*; *Ramakrishnan et al., 2019*; *Xu et al., 2016*).

To achieve the requisite speed and precision of synaptic transmission, the nerve terminals maintain a pool of docked vesicles that can be readily released upon $Ca^{2+}$ influx (*Südhof, 2013*; *Kaeser and Regehr, 2014*; *Südhof and Rothman, 2009*). The prevailing theory is that in a 'release-ready' vesicle, multiple SNARE complexes are firmly held ('clamped') in a partially assembled state. These 'SNAREpins' are then synchronously released by $Ca^{2+}$ to drive fusion dramatically faster than any one SNARE alone (*Rothman et al., 2017*; *Brunger et al., 2018*; *Volynski and Krishnakumar, 2018*). Several lines of evidence suggest that two synaptic proteins, Synaptotagmin-1 (Syt1) and Complexin (Cpx) play a critical role in establishing $Ca^{2+}$-regulated neurotransmitter release (*Geppert et al., 1994*; *Xu et al., 2007*; *Bacaj et al., 2013*; *Yang et al., 2013*; *Huntwork and Littleton, 2007*).

Synaptotagmin-1 is a SV-localized protein with a large cytoplasmic part containing tandem C2A and C2B domains that bind $Ca^{2+}$ (*Fuson et al., 2007*; *Sutton et al., 1995*). It is well established that fast AP-evoked synchronous release is triggered by $Ca^{2+}$ binding to Syt1 C2 domains (*Brose et al., 1992*; *Geppert et al., 1994*; *Littleton et al., 1993*). Genetic analysis shows that Syt1 also plays a crucial role in 'clamping' spontaneous release and $Ca^{2+}$-evoked asynchronous release to ensure high fidelity of the $Ca^{2+}$-coupled synchronous transmitter release (*Xu et al., 2009*; *Bacaj et al., 2013*; *Littleton et al., 1993*). Recently it has been demonstrated that C2B-driven self-oligomerization of Syt1 provides the structural basis for its clamping function (*Wang et al., 2014*; *Bello et al., 2018*; *Tagliatti et al., 2019*). Syt1 is also involved in initial stages of docking of SVs to the plasma membrane (PM), in part mediated by its interaction with the anionic lipids, phosphatidylserine (PS) and phosphatidylinositol 4, 5-bisphosphate (PIP2) on the PM (*Honigmann et al., 2013*; *Parisotto et al., 2012*; *Park et al., 2012*). Besides the membrane interaction, Syt1 also binds the neuronal t-SNAREs on the PM, both independently (primary binding site) and in conjunction with Complexin (tripartite binding site) (*Zhou et al., 2015*; *Zhou et al., 2017*; *Grushin et al., 2019*). The Syt1-SNARE interactions are important for Syt1 role in SV docking, clamping fusion and triggering $Ca^{2+}$-dependent neurotransmitter release (*Zhou et al., 2015*; *Zhou et al., 2017*).

Complexin is a cytosolic α-helical protein that binds and regulates SNARE assembly (*Chen et al., 2002*; *Kümmel et al., 2011*; *Li et al., 2011*). Biochemical analyses reveal that Cpx catalyzes the initial stages of SNARE assembly, but then blocks complete assembly (*Giraudo et al., 2009*; *Kümmel et al., 2011*; *Li et al., 2011*). Physiological studies show that Cpx indeed clamps spontaneous fusion in invertebrate neurons (*Huntwork and Littleton, 2007*; *Wragg et al., 2013*; *Cho et al., 2014*), but its importance as a fusion clamp in mammalian neurons is still under debate (*Yang et al., 2013*; *López-Murcia et al., 2019*; *Courtney et al., 2019*). However, under all conditions, Cpx has been shown to facilitate vesicle priming and promote AP-evoked synchronous release (*Yang et al., 2013*; *Yang et al., 2015*; *López-Murcia et al., 2019*).

It is broadly accepted that Syt1 and Cpx are both involved in clamping spurious or delayed fusion events and synchronize neurotransmitter release to $Ca^{2+}$-influx. However, there is presently no coherent understanding how these proteins assemble and operate together. To gain mechanistic insights into this process, a reductionist approach where the variables are limited, and components can be rigorously controlled is required. We recently described a biochemically-defined fusion system based on a pore-spanning lipid bilayer setup that is ideally suited for this purpose (*Ramakrishnan et al., 2018*). This reconstituted setup is capable of precisely tracking individual vesicle docking, clamping (delay from docking to spontaneous fusion) and $Ca^{2+}$ triggered release at tens of milliseconds timescale (*Coleman et al., 2018*; *Ramakrishnan et al., 2019*). Critically, this setup allows us to examine these discrete sub-steps in the vesicular exocytosis process, independent of alterations in the preceding or following stages (*Coleman et al., 2018*; *Ramakrishnan et al., 2019*; *Ramakrishnan et al., 2018*).

Using the in vitro fusion system, we recently reported that under artificially low-VAMP2 conditions (~13 copies of VAMP2 and ~25 copies of Syt1 per vesicle), Syt1 oligomerization is both necessary and sufficient to establish a $Ca^{2+}$-sensitive fusion clamp (*Ramakrishnan et al., 2019*). Here we extend this study to more physiologically-relevant conditions, using SV mimics reconstituted with ~25 copies of Syt1 and ~70 copies of VAMP2. We report that under these conditions, both Cpx and Syt1 oligomers are needed to stably clamp all SNARE complexes and the reversal of the Syt1 clamp is sufficient to achieve fast, $Ca^{2+}$-triggered synchronized fusion.

## Results

### Synaptotagmin and complexin co-operate to clamp vesicle fusion

With the goal of approximating the physiological context, we chose a reconstitution condition for small unilamellar vesicles (SUV) resulting in an average of 74 copies and 25 copies of outward-facing VAMP2 and Syt1, respectively (*Figure 1—figure supplement 1*). We employed pre-formed t-SNAREs (1:1 complex of Syntaxin1 and SNAP-25) in the planar bilayers (containing 15% PS and 3% PIP2) to both simplify the experimental approach and to bypass the requirement of SNARE-assembling chaperones, Munc18 and Munc13 (*Baker and Hughson, 2016*; *Rizo, 2018*). In most of the

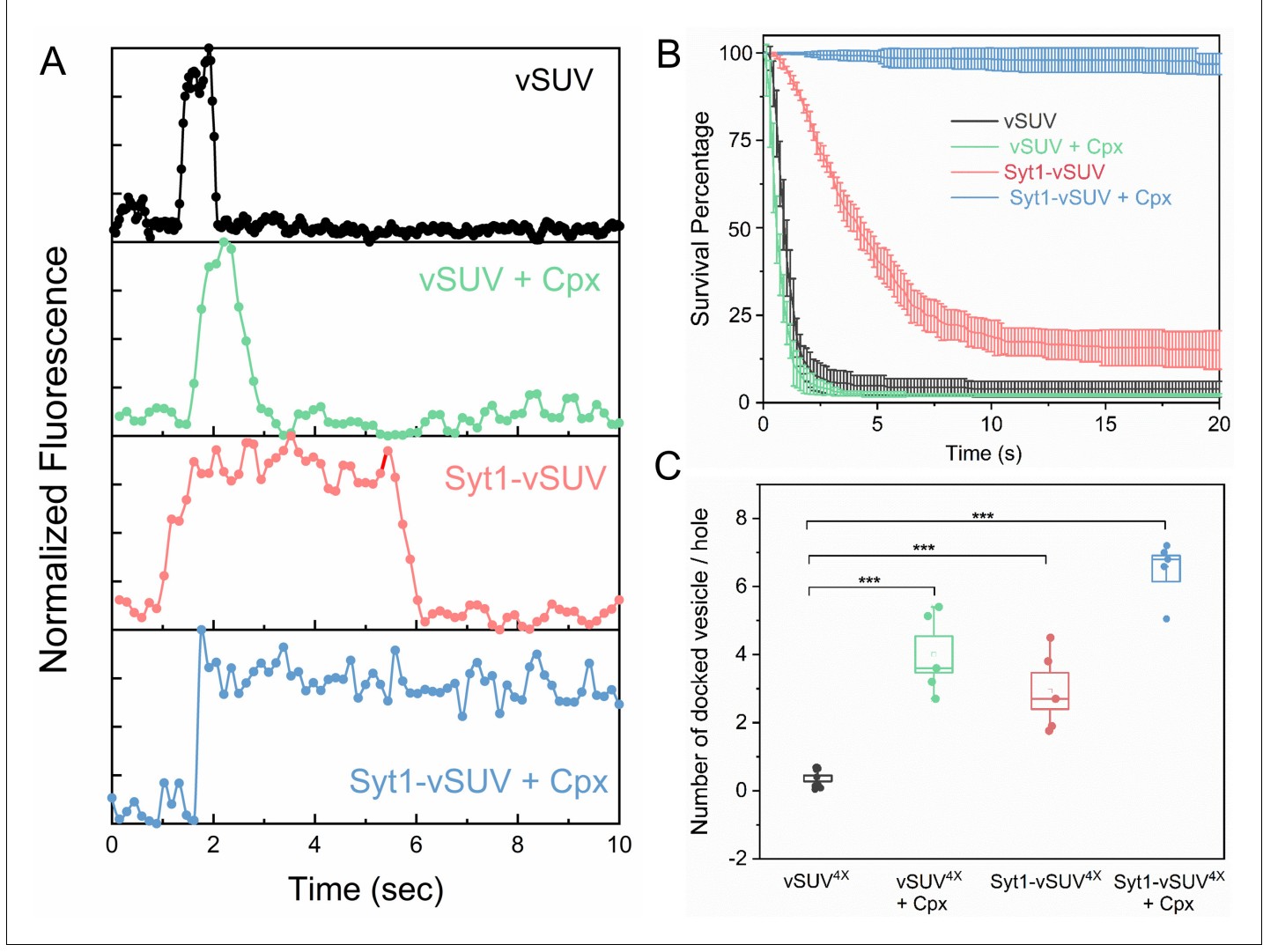

**Figure 1.** Syt1 and Cpx co-operatively clamp SNARE-mediated vesicle fusion under physiologically-relevant conditions. The effect of Syt1 and Cpx on SNARE-driven fusion was assessed using a single-vesicle docking and fusion analysis with a pore-spanning bilayer setup (*Ramakrishnan et al., 2019*; *Ramakrishnan et al., 2018*). (**A**) Representative fluorescence (ATTO647N-PE) traces showing the behavior of small unilamellar vesicles (SUV) containing VAMP2 (vSUV) or Syt1 and VAMP2 (Syt1-vSUV) on t-SNARE containing bilayer in the presence or the absence of Cpx. (**B**) The time between docking and fusion was measured for each docked vesicle and the results for the whole population are presented as a survival curve. vSUVs (black curve) are diffusively mobile upon docking (*Figure 1—figure supplement 2*) and fuse spontaneous with a half-time of ~1 s. Addition of soluble Cpx (2 µM) does not change this behavior (green curve). Inclusion of Syt1 in the v-SUV (red curve) does not block fusion but increases the time from docking-to-fusion (~5 s half-life), in effect delaying the kinetics of fusion. When included together Syt1 and Cpx (blue curves) fully arrest fusion to produce stably docked SUVs that attach and remain in place during the entire period of observation. (**C**) Syt1 and Cpx, both individually and collectively increase the number of docked vesicles. In all cases, a mutant form of VAMP2 (VAMP2$^{4X}$) which eliminated fusion was used to unambiguously estimate the number of docked vesicles after the 10 min interaction phase. The average values and standard deviations from three to four independent experiments are shown for each condition. In sum, 500–1000 vesicles were analyzed for each condition.

The online version of this article includes the following figure supplement(s) for figure 1:

**Figure supplement 1.** Coomaisse-stained SDS-PAGE analysis of the proteins used in this study.

**Figure supplement 2.** Representative time-lapse fluorescence (ATTO647N-PE) images showing the behavior of VAMP2- containing SUVs in presence of Syt1 and/or Cpx.

**Figure supplement 3.** Survival analysis showing that vSUVs in the presence of Syt1 and Cpx dock and remain stably clamped for up to 1 hr*.

experiments, we used fluorescently-labelled lipid (2% ATTO647-PE) included in the SUVs to track the docking, diffusion and fusion of individual SUVs (*Figure 1A*).

**Table 1.** Survival probabilities at specific points in time post-docking (Kaplan Meier estimators) for the VAMP2-containing vesicles (vSUV) in the presence of Cpx, Syt1 or both. The corresponding survival curves are shown in *Figure 1B*

| Time (s) (post-docking) | vSUV | vSUV + Cpx | Syt1-vSUV | Syt1-vSUV + Cpx |
|---|---|---|---|---|
| 0.588 | 0.7978 | 0.4945 | 0.9915 | 1.0000 |
| 1.029 | 0.4134 | 0.2277 | 0.9519 | 0.9989 |
| 2.058 | 0.0941 | 0.0544 | 0.8045 | 0.9963 |
| 3.087 | 0.0583 | 0.0319 | 0.6361 | 0.9936 |
| 4.116 | 0.0491 | 0.0267 | 0.5110 | 0.9917 |
| 5.145 | 0.0439 | 0.0243 | 0.3984 | 0.9888 |
| 7.497 | 0.0421 | 0.0215 | 0.2482 | 0.9821 |
| 9.996 | 0.0388 | 0.0211 | 0.1853 | 0.9814 |
| 15.141 | 0.0381 | 0.0209 | 0.1547 | 0.9794 |
| 20.139 | 0.0379 | 0.0201 | 0.1505 | 0.9682 |

We initially focused on the kinetics of constitutive fusion to assess the ability of Syt1 and Cpx to 'clamp' SNARE-driven fusion in the absence of $Ca^{2+}$. We monitored large ensembles of vesicles to determine the percent remaining unfused as a function of time elapsed after docking and quantified as 'survival percentages' (*Figure 1B*). Docked immobile vesicles that remained un-fused during the initial 10 min observation period were defined as 'clamped' (*Ramakrishnan et al., 2019*). Vesicles containing VAMP2 only (vSUV) that docked to the t-SNARE containing bilayer surface were mobile and a majority (>95%) spontaneously fused typically with a $t_{1/2}$ ~ 1 sec post-docking (*Figure 1B*, *Figure 1—figure supplement 2*, *Table 1*, *Table 2*, *Video 1*).

Inclusion of wild-type Syt1 in the vesicles (Syt1-vSUVs) enhanced the vesicle docking rate, with ~8 fold increase in total number of docked vesicles (*Figure 1C*). The majority (~80%) of docked Syt1-vSUVs remained mobile on the bilayer surface and fused on an average ~5–6 s after docking (*Figure 1B*, *Figure 1—figure supplement 2*, *Table 1*, *Table 2*, *Video 2*). The remaining small fraction (~20%) were immobile and stably clamped. This is in stark contrast to our earlier finding under low-copy VAMP2 conditions wherein the bulk of the Syt1-vSUVs (>90%) were stably clamped (*Ramakrishnan et al., 2019*). The pronounced docking-to-fusion delay introduced by Syt1 ($t_{1/2}$ ~5 sec for Syt1-vSUV compared to ~1 s for vSUV) suggests that under physiologically-relevant ('normal' VAMP copy number) conditions, Syt1 alone can meaningfully delay but not stably clamp fusion.

This unstable Syt1 clamp was stabilized by addition of Cpx (*Figure 1B*, *Figure 1—figure supplement 2*, *Video 3*). In the presence of 2 μM of soluble Cpx, all Syt1-vSUVs were immobile following docking (*Figure 1B*), and they rarely fused over the initial observation period (*Figure 1B*, *Figure 1—figure supplement 2*, *Table 1*, *Table 2*, *Video 3*). In fact, these vesicles remained stably docked up to 1 hr without fusing (*Figure 1—figure supplement 3*). Furthermore, Syt1 and Cpx together significantly increased (~18 fold) the total number of docked vesicles (*Figure 1C*). Thus, we find that Syt1 and Cpx act synergistically to increase the rate of vesicle docking and serve to effectively block fusion of the docked vesicles under resting conditions.

Addition of soluble Cpx (2 μM) alone produced a ~10 fold increase in the number of docked vesicles (*Figure 1C*) but did not change the behavior of the docked vSUVs (*Figure 1B*, *Figure 1—*

**Table 2.** The survival curves in *Figure 1B* were compared pair-wise using the log-rank test to determine the statistical significance (p-values) of the observed effects of Syt1 and Cpx on v-SUV fusion

| | vSUV | vSUV + cpx | Syt1-vSUV | Syt1-vSUV + cpx |
|---|---|---|---|---|
| vSUV | n/a | p=0.731 | p<0.001 | p<0.0001 |
| vSUV + Cpx | p=0.731 | n/a | p<0.0001 | p<0.0001 |
| Syt1-vSUV | p<0.001 | p<0.001 | n/a | p<0.001 |
| Syt1-vSUV + Cpx | p<0.0001 | p<0.0001 | p<0.001 | n/a |

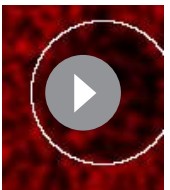

**Video 1.** Representative video showing the docking, diffusion and spontaneous fusion of a vSUV to free-standing t-SNARE containing lipid bilayer containing t-SNARE complex. The movie was acquired using a Leica confocal scanning microscope at a speed of 7 frames/sec and the ATTO647N-PE fluorescence was used to track the fate of the vesicle. For clarity, a single hole corresponding to 5 μm suspended bilayer marked by the white circle is shown.

https://elifesciences.org/articles/54506#video1

figure supplement 2, *Table 1*, *Table 2*, *Video 4*). In the presence of Cpx alone, virtually all docked vSUVs fused spontaneously typically within 1–2 s (*Figure 1A,B*). This meant that Syt1 somehow synergizes with Cpx to form the overall fusion clamp.

## Synaptotagmin and complexin establish fast, $Ca^{2+}$-triggered vesicle fusion

We then investigated the effect of $Ca^{2+}$ on the stably Syt1/Cpx-clamped vesicles (*Figure 2*). We estimated the time of arrival of $Ca^{2+}$ at/near the docked vesicles using a lipid-conjugated $Ca^{2+}$ indicator (Calcium green C24) attached to the planar bilayer (*Figure 2—figure supplement 1*). Influx of free $Ca^{2+}$ (1 mM) triggered simultaneous fusion of >90% of the docked vesicles (*Figure 2B*, *Video 5*). These vesicles fused rapidly and synchronously, with a characteristic time-constant (τ) of ~110 msec following the arrival of $Ca^{2+}$ locally (*Figure 2C*). This estimate is constrained by the temporal resolution limit (150 msec per frame) of our imaging experiment. Indeed, most of $Ca^{2+}$-triggered fusion occurs within a single frame (*Figure 2C*). We thus suspect that the true $Ca^{2+}$-driven fusion rate is <100 msec. Notably, the Syt1/Cpx clamped vesicles remained $Ca^{2+}$-sensitive even 1 hr post-docking (*Figure 1—figure supplement 3*). Our data indicate that Syt1 and Cpx acting together *synchronize* vesicle fusion to $Ca^{2+}$ influx and greatly *accelerate* the underlying SNARE-mediated fusion process (which typically occurs at a rate of ~1 s). We also tested and confirmed these findings with a content-release assay using sulforhodamine B-loaded SUVs (*Figure 2—figure supplement 2*) under similar experimental conditions. Overall, we find that Syt1 and Cpx act co-operatively to clamp the SNARE assembly process to generate and maintain a pool of docked vesicles that can be triggered to fuse rapidly and synchronously upon $Ca^{2+}$ influx.

## Synaptotagmin and complexin clamp different sets of SNARE complexes

We next examined if Syt1 and Cpx act on the same SNARE complexes sequentially or if they function separately to produce molecularly-distinct clamped SNAREpins under the same docked vesicles. To this end, we employed an accessibility-dependent competition assay (*Figure 3*). We washed Cpx out from stably clamped Syt1/Cpx vesicles (by dilution) in the absence or the presence of excess inhibitory soluble cytoplasmic domain of the t-SNARE complex (CDT). Cpx binds to half-zippered (clamped) SNAREpins with a $K_d$ ~0.5 μM (*Krishnakumar et al., 2011*; *Kümmel et al., 2011*) and is therefore expected to freely dissociate when the bulk concentration of CPX is reduced well below that level. CDT will bind and sequester/inactivate any free VAMP2 on the vesicles and is also expected to effectively compete out bilayer-anchored t-SNAREs. This is because CDT can form fully-zippered SNARE complexes (stabilized by ~70 $k_B$T) as compared to the half-zippered SNAREpins (~35 $k_B$T) formed by bilayer-anchored t-SNAREs (*Li et al., 2016*). We reasoned that if Syt1 and Cpx act on the same SNARE complex, then CDT treatment (with Cpx wash-out) would irreversibly block all vesicle fusion. However, if Syt1 and Cpx clamp different SNARE complexes, then some SNAREpins might be sequestered/protected from CDT by Syt1, thereby keeping the vesicles clamped yet sensitive to $Ca^{2+}$ influx.

We used fluorescently-labeled Cpx to test and confirm the near-complete washout of Cpx from the clamped Syt1/Cpx vesicles following the extensive (40X) buffer wash (*Figure 3—figure supplement 1*). Without CDT, the docked vesicles proceeded to fuse spontaneously following the buffer wash (*Figure 3A*, *Figure 3—figure supplement 2*). This further confirmed that both Syt1 and Cpx are needed to produce a stably clamped state. In the presence of CDT, most of the vesicles remained docked even after the removal of previously-bound Cpx by the buffer wash (*Figure 3B*,

*Figure 3—figure supplement 2*). Subsequent addition of $Ca^{2+}$ (1 mM) triggered rapid and synchronous fusion of the docked vesicles (*Figure 3B*, *Figure 3—figure supplement 2*), with fusion kinetics similar to the control experiments (*Figure 2C*). This implied that there are at least two types of clamped SNAREpins under a docked vesicle – those clamped by Syt1 (which are shielded from CDT) and others arrested by Cpx (which become accessible to CDT following the buffer wash-out). It further indicated that even though both Syt1 and Cpx are required to produce a stably 'clamped' vesicle, the activation of the Syt1-associated SNAREpins is sufficient to elicit rapid, $Ca^{2+}$-synchronized vesicular fusion.

## Molecular mechanism of synaptotagmin clamp and $Ca^{2+}$ activation of fusion

Considering these findings, we sought to establish the molecular determinants of the Syt1 clamp and its reversal by $Ca^{2+}$. To focus on the Syt1 component of the clamp, we tested the effect of specific Syt1 mutations using low copy VAMP2 conditions, i.e. SUVs containing ~13 copies of VAMP2 and ~25 copies of Syt1 (wild type or mutants) in the absence of Cpx. Consistent with our earlier report (*Ramakrishnan et al., 2019*), wild-type Syt1 (Syt1^WT) alone was sufficient to produce stably-clamped vesicles under these conditions (*Figure 4A*, *Table 3*, *Table 4*). Selective disruption of Syt1-SNARE 'primary' binding using the previously described (*Zhou et al., 2015*; *Zhou et al., 2017*) mutations in Syt1 C2B domain (R281A/E295A/Y338W/R398A/R399A; Syt1^Q) and t-SNARE SNAP25 (K40A/D51A/E52A/E55A/D166A; SNARE^Q) abolished the Syt1 clamp (*Figure 4A*, *Table 3*, *Table 4*,), with >99% of the docked Syt1^Q vesicles fusing constitutively in the 10 min observation period (Note: From this point onwards, the 'primary' site mutation is simply referred as Syt1^Q). On the other hand, Syt1 mutations (L387Q/L394Q; Syt1^LLQQ) that disrupt the hydrophobic interaction that is an integral part of the SNARE-Cpx-Syt1 'tripartite' interface (*Zhou et al., 2017*) had no effect on the Syt1 clamp (*Figure 4A*, *Table 3*, *Table 4*). Destabilization of the Syt1 C2B oligomers with a point-mutation (F349A)(*Bello et al., 2018*) also abrogated the Syt1 clamp, wherein ~ 85% of docked Syt1^F349A vesicles proceeded to fuse spontaneously (*Figure 4A*, *Table 3*, *Table 4*). However, disrupting $Ca^{2+}$ binding to the C2B domain (D309A, D363A, D365A; Syt1^DA) (*Shao et al., 1996*) had no effect on Syt1 clamping function, with all vesicles remaining un-fused (*Figure 4A*, *Table 3*, *Table 4*). This meant that Syt1 ability to oligomerize and bind SNAREpins via the primary binding site is key to its clamping function.

Addition of $Ca^{2+}$ (1 mM) triggered rapid and synchronous fusion of all of the docked Syt1^WT and Syt1^LLQQ vesicles and the remaining minority fraction (~15%) of the 'clamped' Syt1^F349A containing vesicles (*Figure 4B*). In fact, the docked Syt1^F349A vesicles were indistinguishable in their behavior from Syt1^WT vesicles, suggesting the Syt1 oligomerization is not critical for the $Ca^{2+}$-activation mechanism. In contrast, docked vesicles containing the $Ca^{2+}$-binding mutant (Syt1^DA) never fused even after $Ca^{2+}$-addition (*Figure 4B*). Similarly, $Ca^{2+}$-

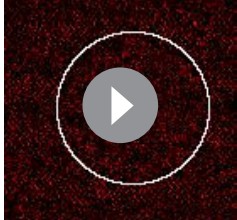

**Video 2.** Representative video showing the docking, slow diffusion and delayed spontaneous fusion of a Syt1- containing vSUV to free-standing lipid bilayer containing t-SNAREs. The movie was acquired using a Leica confocal scanning microscope at a speed of 7 frames/sec and the ATTO647N-PE fluorescence was used to track the vesicle. For clarity, a single hole corresponding to 5 μm suspended bilayer marked by the white circle is shown.
https://elifesciences.org/articles/54506#video2

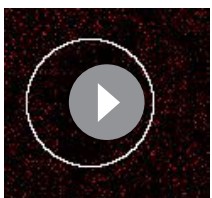

**Video 3.** Representative video showing the immobile docking of a Syt1-vSUV to a free-standing t-SNARE containing lipid bilayer in the presence of 2 μM Cpx. The ATTO647N-DOPE introduced in the SUV was used to track the vesicle and the movie was acquired using a Leica confocal scanning microscope at a speed of 7 frames/sec. For clarity, a single hole corresponding to 5 μm suspended bilayer marked by the white circle is shown.
https://elifesciences.org/articles/54506#video3

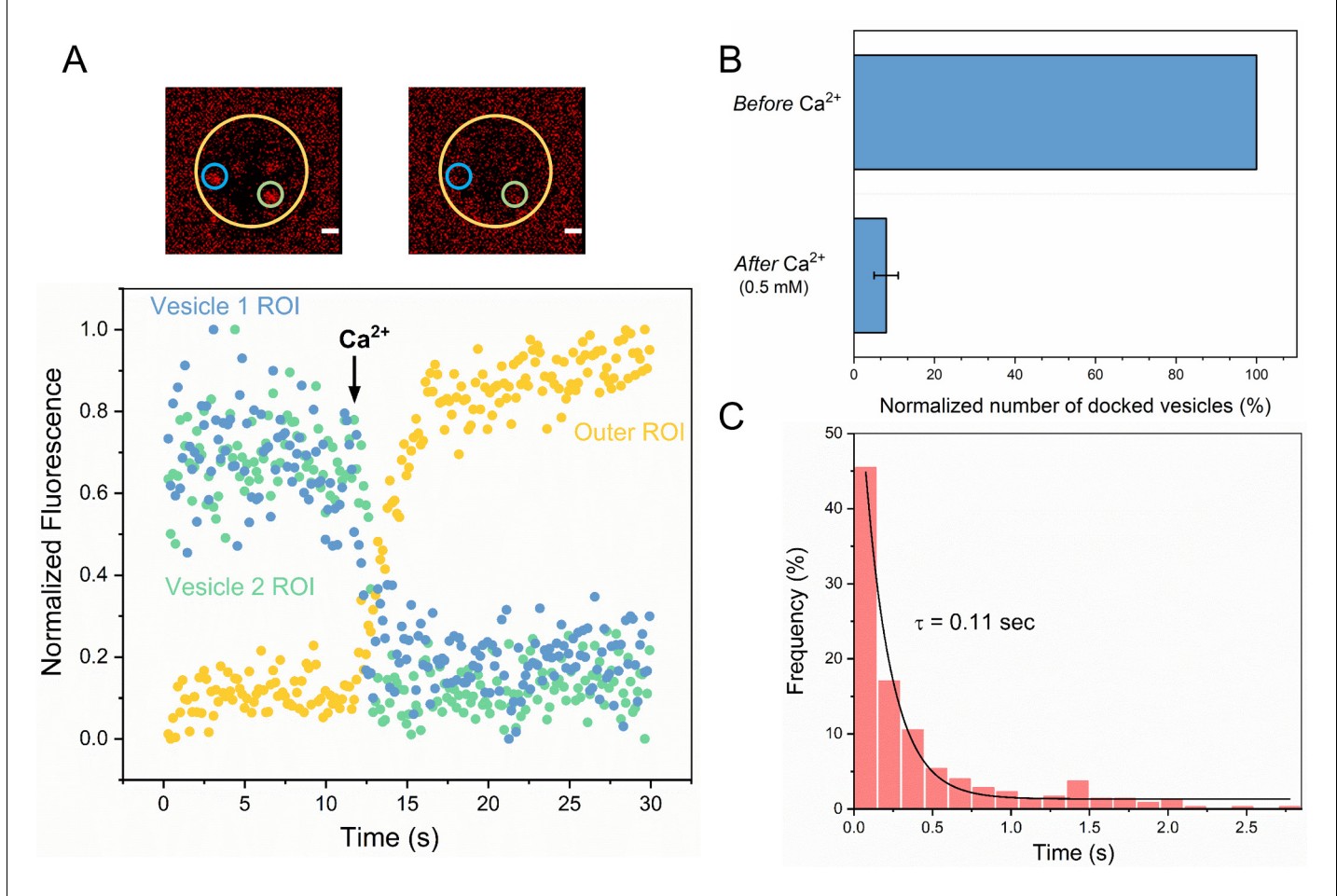

**Figure 2.** Syt1/Cpx clamped vesicles fuse synchronously and rapidly following $Ca^{2+}$-addition. (**A**) Representative fluorescence images (top) and quantitation of change in fluorescence signal (bottom) before and after addition of 1 mM $Ca^{2+}$ shows that vesicles clamped by Syt1/Cpx are sensitive to $Ca^{2+}$. Fusion was attested by a burst and sudden decrease in fluorescence (ATTO647N-PE) intensity as the lipids diffuse away. To visualize this, fluorescence was simultaneously monitored in a circular region of interest (ROI) encompassing the docked vesicle (vesicle ROI, green and blue circles) and in a surrounding annular ROI (outer ROI, yellow circle). Corresponding to actual fusion events, we observed a sudden decrease of fluorescence intensity in the vesicle ROI with a concomitant increase of fluorescence in the annular outer ROI. Note that the two docked vesicles fuse synchronously in response to $Ca^{2+}$-influx. (**B**) End-point analysis at 1 min post $Ca^{2+}$-addition shows that >90% of all clamped vesicles fuse following $Ca^{2+}$ addition. (**C**) Kinetic analysis shows that the Syt1/Cpx clamped vesicles fuse rapidly following $Ca^{2+}$-addition with a characteristic time constant of 0.11 s. This represents the temporal resolution limit of our recordings and the true $Ca^{2+}$-triggered fusion rate is likely well below 0.1 s. Thus, Syt1 and Cpx synchronize vesicle fusion to $Ca^{2+}$-influx and accelerate the underlying fusion process. The average values and standard deviations from three independent experiments (with ~1000 vesicles in total) is shown.

The online version of this article includes the following figure supplement(s) for figure 2:

**Figure supplement 1.** $Ca^{2+}$-sensor dye, Calcium Green, introduced in the suspended bilayer (via a lipophilic 24-carbon alkyl chain) was used to monitor the arrival of $Ca^{2+}$ at/near the docked vesicles.

**Figure supplement 2.** The effect of Syt1 and Cpx on SNARE-driven fusion assessed using a content-release assay with Sulforhodamine B loaded vesicles.

influx failed to trigger the fusion of the residual (~1%) Syt1$^Q$ vesicles. However, the relatively small number of docked Syt1$^Q$ prior to $Ca^{2+}$-influx precludes any meaningful quantitative analysis. Nonetheless, our data suggest that $Ca^{2+}$-binding to the Syt1 C2B domain and its simultaneous interaction with the t-SNARE protein via the primary binding site is required for $Ca^{2+}$-triggered reversal of the fusion clamp.

We also tested the effect of the Syt1 mutants using vesicles containing physiological VAMP2 and Syt1 copy numbers in the presence of 2 µM Cpx (*Figure 5A*, *Table 5*, *Table 6*). For all mutations

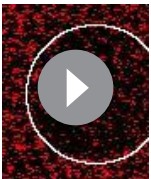

**Video 4.** Representative video showing the docking, diffusion and spontaneous fusion of a vSUV to free-standing t-SNARE containing lipid bilayer in the presence of 2 μM Cpx. The movie was acquireusing a Leica confocal scanning microscope at a speed of 7 frames/sec and the ATTO647N-PE fluorescence was used to track the vesicle. For clarity, a single hole corresponding to 5 μm suspended bilayer marked by the white circle is shown.

https://elifesciences.org/articles/54506#video4

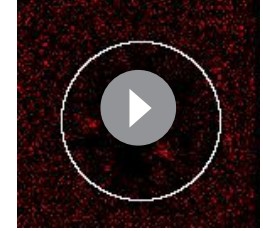

**Video 5.** Representative video showing the rapid and synchronous fusion of multiple Syt1/Cpx clamped vesicles to free-standing t-SNARE containing lipid bilayer triggered by $Ca^{2+}$-influx (1 mM free). The ATTO647N-DOPE introduced in the SUV was used to track the fate of the vesicle and the movie was acquired using a Leica confocal scanning microscope at a speed of 7 frames/sec. For clarity, a single hole corresponding to 5 μm suspended bilayer marked by the white circle is shown.

https://elifesciences.org/articles/54506#video5

tested, we observed immobile, stably docked vesicles (*Figure 5A*, *Table 5*, *Table 6*). Surprisingly, the Cpx-dependent clamp was observed even under conditions wherein the Syt1 clamp was absent (i.e. Syt1$^Q$ and Syt1$^{F349A}$). Survival analysis revealed that both Syt1$^Q$ and Syt1$^{F349A}$ mutants introduce a meaningful delay in the overall fusion process albeit less than that observed with a stable Syt1 clamp (*Figure 5—figure supplement 1*). This implies that the kinetic delay introduced by Syt1, independent of its ability to clamp, is sufficient to enable Cpx to function as a fusion clamp, perhaps by providing time for Cpx to bind and block all SNAREpins.

However, the clamped Syt1$^Q$ and Syt1$^{F349A}$ vesicles were insensitive to $Ca^{2+}$ and did not fuse following $Ca^{2+}$ (1 mM) addition as opposed to the rapid and synchronous fusion observed with the majority of the Syt1$^{WT}$ and Syt1$^{LLQQ}$ vesicles (*Figure 5B*). Notably, a significant partial fraction (~25%) of the Syt1$^{LLQQ}$ vesicles remained un-fused even following $Ca^{2+}$ addition (*Figure 5B*). This suggests that while the Syt1-Cpx-SNARE tripartite interface is not essential for establishing the fusion clamp, it is likely important in the $Ca^{2+}$-triggering of fusion.

Taken together, our data indicate that (i) the Syt1 clamp and associated SNAREpins are critical for $Ca^{2+}$-triggered fusion and (ii) Cpx, on its own, irreversibly blocks vesicle fusion. These conclusions are corroborated by Cpx washout (±CDT) experiments on the Syt1$^Q$ vesicles (*Figure 5—figure supplement 2*). Without CDT, Syt1$^Q$ vesicles fused spontaneously following the Cpx washout. In the presence of CDT, the Syt1$^Q$ vesicles remained docked following the buffer wash, but were insensitive to $Ca^{2+}$ and failed to fuse following the addition of 1 mM $Ca^{2+}$ (*Figure 5—figure supplement 2*). This denotes that in Syt1$^Q$ vesicles (and presumably in Syt1$^{F349A}$ vesicles), all SNAREpins are clamped by Cpx alone and become accessible to CDT block following the Cpx wash-out.

## Discussion

Here we report that Syt1 and Cpx act concomitantly to clamp SNARE-driven constitutive fusion events (*Figure 1*). We find there are at least two distinct sets of clamped SNAREpins under every docked vesicle – a small population that is reversibly clamped by Syt1 oligomers and the remainder that is irreversibly blocked by Cpx. (*Figure 3*). These results taken together with the known structural and biochemical properties of Syt1 and Cpx prompts a novel 'synergistic clamping' mechanism.

We posit that the Syt1 C2B domain binds PIP2 (via the poly-lysine motif) on the PM and assembles into ring-like oligomeric structures (*Bello et al., 2018*; *Wang et al., 2014*; *Zanetti et al., 2016*). The Syt1 oligomers concurrently bind the t-SNAREs via the 'primary' interface (*Zhou et al., 2015*; *Zhou et al., 2017*; *Grushin et al., 2019*). The Syt1-t-SNARE interaction, which likely precedes the engagement of the v- and the t-SNAREs, positions the Syt1 such that it sterically blocks the full assembly of the associated SNAREpins (*Grushin et al., 2019*). The Syt1 oligomers in addition to creating a stable steric impediment could also radially restrain the assembling SNAREpins

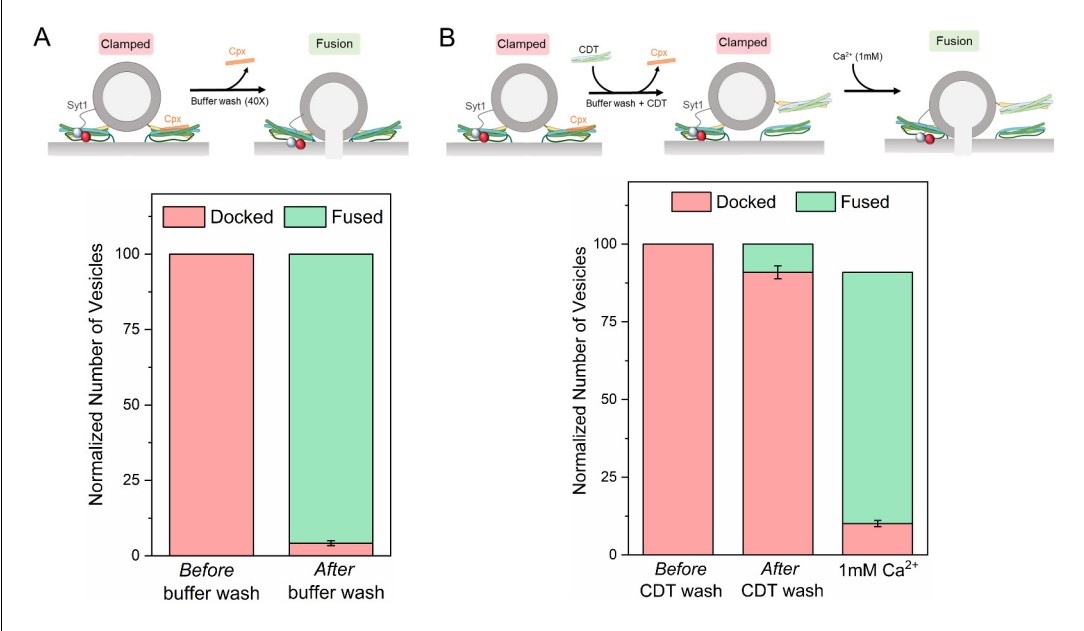

**Figure 3.** Syt1 and Cpx bind and clamp different pools of SNAREpins. (**A**) In situ removal of Cpx from the Syt1/Cpx clamped state by extensive (40X) buffer wash triggers spontaneous fusion of the docked vesicles. This further confirms that both Syt1 and Cpx are required to produce a stable clamped state. (**B**) Inclusion of soluble cytoplasmic domain of t-SNAREs (CDT) blocked the spontaneous fusion events triggered by elimination of Cpx from the Syt1/Cpx clamped vesicles. The CDT-treated vesicles remain sensitive to $Ca^{2+}$ influx and most of the vesicles fuse rapidly and synchronously following the addition of 1 mM $Ca^{2+}$. This indicates that a sub-set of SNAREpins are protected against CDT even in the absence of Cpx, implying that Syt1 and Cpx likely engage and clamp different set of SNAREpins. It further shows that the Syt1-associated SNAREpins are sufficient to catalyze rapid $Ca^{2+}$-triggered vesicle fusion. Data (average ± standard deviation) obtained from four to five independent experiments with at least 200 vesicles in total are shown. Note: Only single SNAREpins clamped by either Syt1 or Cpx is shown for illustrative purposes.

The online version of this article includes the following figure supplement(s) for figure 3:

**Figure supplement 1.** Representative fluorescence images showing that the extensive (40X) buffer wash results in complete washout of Cpx from clamped Syt1/Cpx vesicles.

**Figure supplement 2.** Respresentative fluorescence images showing the effect of CDT wash and subsequent $Ca^{2+}$ addition.

(*Grushin et al., 2019*; *Rothman et al., 2017*). It is worth noting in this configuration the helical extension of Syt1 C2B that forms the 'tripartite' interface with Cpx and SNAREs contacts the PM and is thus unavailable for tripartite binding (*Rothman et al., 2017*; *Volynski and Krishnakumar, 2018*; *Grushin et al., 2019*).

The Syt1 oligomers can bind and clamp only a small sub-set of SNAREpins (which we refer to as 'central SNAREpins') as the number of potential SNAREpins per SV far exceeds the Syt1 density (~70 copies of VAMP2 vs. ~20 copies of Syt1). We suggest that the remaining 'peripheral' SNAREs would then be unrestrained and can fully zipper to catalyze vesicle fusion, though at an impeded rate (*Figure 1*). It is tempting to speculate that the limited set of Syt1-clamped 'central' SNAREpins correspond physiologically to the six symmetrically organized protein densities that underlie each docked synaptic-like vesicle visualized by cryo-electron tomography analysis (*Li et al., 2019*). Indeed, this symmetrical organization depends on Syt1 oligomerization (*Li et al., 2019*).

Cpx potentially binds the SNARE complex only when the Syntaxin and VAMP2 are partially-assembled (*Chen et al., 2002*; *Kümmel et al., 2011*) and competitively blocks the complete zippering of the C-terminal portion of the VAMP2 SNARE motif (*Kümmel et al., 2011*; *Li et al., 2011*; *Giraudo et al., 2009*). It is well-established that the C-terminal portion of VAMP2 assembles with extraordinarily high energy and rate acting as the major power stroke to drive membrane fusion (*Gao et al., 2012*). Consequently, Cpx, on its own, is ineffective in clamping SNARE-driven vesicle fusion (*Figure 1*).

We envision that under physiological conditions, Syt1 sets the stage for Cpx to bind and clamp the 'peripheral' SNARE complexes. Syt1, in addition to fully-arresting the central SNAREpins, also

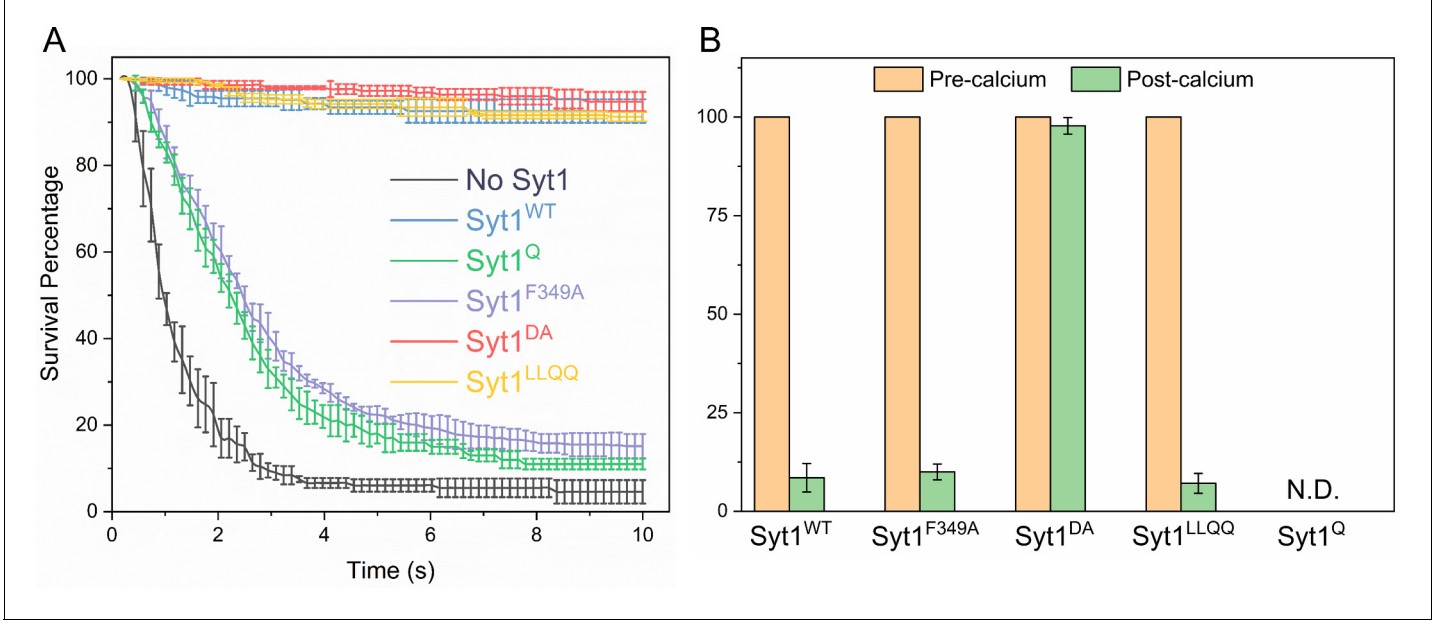

**Figure 4.** Molecular determinants of Syt1 clamp and its reversal by $Ca^{2+}$. To focus on the Syt1 component of the fusion clamp, all fusion analysis was carried out using vesicles containing low copy VAMP2 (~13 copies) with normal number (~25 copies) of Syt1 molecules (wild-type or targeted mutations) in the absence of Cpx. (**A**) Survival analysis shows that disrupting the Syt1-SNARE primary interface (Syt1$^Q$, green curve) or destabilizing Syt1 oligomerization (Syt1$^{F349A}$, purple curve) abrogates the Syt1 clamp, whilst the Syt1-Cpx-SNARE tripartite interface (Syt1$^{LLQQ}$, yellow curve) and the $Ca^{2+}$-binding motif on the Syt1 C2B domain (Syt1$^{DA}$, red curve) are not involved in establishing the fusion clamp. (**B**) Addition of $Ca^{2+}$ triggered rapid fusion of the majority (>90%) of Syt1$^{LLQQ}$ and the remainder (~15%) of the docked Syt1$^{F349A}$ vesicles, very similar to the behavior of the Syt1$^{WT}$ vesicles. Predictably, blocking $Ca^{2+}$-binding to C2B domain rendered the vesicle $Ca^{2+}$-insensitive, with the majority of Syt1$^{DA}$ remaining un-fused. We did not have sufficient number of docked Syt1$^Q$ vesicles to do a quantitative analysis, but qualitatively, the few that remained docked failed to fuse following $Ca^{2+}$-addition. This implies that both C2B binding to $Ca^{2+}$ and SNAREs are required for the $Ca^{2+}$-activation, but the ability to form oligomers or Syt1-Cpx-SNARE tripartite interface are not crucial for the $Ca^{2+}$-triggered reversal of the Syt1 clamp. The average values and standard deviations from four independent experiments are shown for each condition. In total,~250 vesicles were analyzed for each condition.

kinetically hinders the assembly of the peripheral SNAREpins enabling Cpx to effectually arrest their complete zippering. In this manner, the sequential action of Syt1 and Cpx on different populations of SNAREpins would be involved in the generation of the stably docked, release-ready vesicles. Under each vesicle, Syt1 oligomers bind and clamp a small sub-set of available SNAREpins at the early stages of docking, which in turn enables Cpx to block the remainder of the SNAREpins. This molecular model would readily explain the importance of both Syt1 and Cpx in establishing the

**Table 3.** Survival probabilities at specific points in time post-docking (Kaplan Meier estimators) for the different Syt1 mutants under low-copy VAMP2 conditions in the absence of Cpx. The corresponding survival curves are shown in *Figure 4A*

| Time (s) (post-docking) | vSUV | Syt1$^{WT}$ | Syt1$^{F349A}$ | Syt1$^{3DA}$ | Syt1$^Q$ | Syt1$^{LLQQ}$ |
|---|---|---|---|---|---|---|
| 0.588 | 0.7925 | 0.9965 | 0.9597 | 0.9936 | 0.9612 | 0.9975 |
| 1.029 | 0.4679 | 0.9799 | 0.8951 | 0.9931 | 0.8353 | 0.9963 |
| 2.058 | 0.1697 | 0.9546 | 0.5992 | 0.9857 | 0.5403 | 0.9854 |
| 3.087 | 0.0899 | 0.9546 | 0.3822 | 0.9794 | 0.3193 | 0.9525 |
| 4.116 | 0.0662 | 0.9341 | 0.2745 | 0.9762 | 0.2163 | 0.9416 |
| 5.145 | 0.0606 | 0.9341 | 0.2209 | 0.9722 | 0.1732 | 0.9416 |
| 7.497 | 0.0549 | 0.9255 | 0.1663 | 0.9597 | 0.1253 | 0.9160 |
| 9.996 | 0.0459 | 0.9189 | 0.1514 | 0.9475 | 0.1152 | 0.9124 |

**Table 4.** The survival curves in **Figure 4A** were compared pair-wise using the log-rank test to determine the statistical significance (p-values) of the effects of the Syt1 mutants under low-copy VAMP2 conditions in the absence of Cpx.

| | vSUV | Syt1[WT] | Syt1[F349A] | Syt1[3DA] | Syt1[Q] | Syt1[LLQQ] |
|---|---|---|---|---|---|---|
| vSUV | n/a | p<0.0001 | p<0.004 | p<0.0001 | p<0.009 | p<0.0001 |
| Syt1[WT] | p<0.0001 | n/a | p<0.0001 | p=0.912 | p<0.0001 | p=0.885 |
| Syt1[F349A] | p<0.004 | p<0.0001 | n/a | p<0.0001 | p=0.113 | p<0.0001 |
| Syt1[3DA] | p<0.0001 | p=0.912 | p<0.0001 | n/a | p<0.0001 | p=0.915 |
| Syt1[Q] | p<0.009 | p<0.0001 | p=0.113 | p<0.0001 | n/a | p<0.0001 |
| Syt1[LLQQ] | p<0.0001 | p=0.885 | p<0.0001 | p=0.915 | p<0.0001 | n/a |

fusion clamp as evidenced in several genetic deletion studies (*Bacaj et al., 2013*; *Geppert et al., 1994*; *Cho et al., 2014*; *Martin et al., 2011*; *Littleton et al., 1993*; *Yang et al., 2013*).

We also report that the reversal of the Syt1 clamp is sufficient to drive rapid $Ca^{2+}$-triggered fusion of the docked vesicles (*Figure 2*). It involves Syt1 C2B domain binding both $Ca^{2+}$ and the SNARE complex (*Figure 4*). We have recently demonstrated that $Ca^{2+}$-binding to Syt1 C2 domains induce a large-scale conformational rearrangement of the Syt1-SNARE complex on the lipid membrane surface, disrupting the pre-fusion clamped architecture (*Grushin et al., 2019*). This, in effect, reverses the Syt1 clamp, allowing the associated SNAREs to complete zippering and drive fusion. This implies that only a small fraction of available SNAREpins per vesicle (i.e. only those associated with Syt1) are involved in the $Ca^{2+}$-activation process. This is consistent with the earlier reports that 2–3 SNARE complexes can be sufficient to facilitate $Ca^{2+}$-evoked synchronous neurotransmitter release (*Sinha et al., 2011*; *Mohrmann et al., 2010*). Indeed, recent modeling studies considering the concept of mechanical coupling have predicted that an optimum of 4–6 SNAREpins is required to achieve sub-millisecond vesicular release (*Manca et al., 2019*).

There is a long-standing debate over the role of Cpx in establishing a fusion clamp (*Yang et al., 2013*; *López-Murcia et al., 2019*). We find that Cpx is an integral part of the overall clamping mechanism and Syt1 and Cpx play distinct roles in clamping different pools of SNAREpins. However, Cpx requires a kinetic delay (likely introduced by Syt1) to block vesicle fusion and this inhibition is not released by $Ca^{2+}$. This raises the intriguing possibility that the Cpx clamp is not necessarily reversed during the $Ca^{2+}$-activation process. The molecular details of the observed Cpx 'clamp' and its physiological relevance remains to be determined.

Overall, our data are in alignment with the emerging view that Syt1 plays a pivotal role in orchestrating $Ca^{2+}$-regulated neurotransmitter release. Syt1 functions both as a fusion clamp and the principal $Ca^{2+}$-sensor to establish $Ca^{2+}$-regulation of vesicular fusion (*Geppert et al., 1994*; *Littleton et al., 1993*). Furthermore, Syt1, by virtue of self-assembling into oligomeric structures, also provides the molecular framework to organize the exocytic machinery into a co-operative

**Table 5.** Survival probabilities at specific points in time post-docking (Kaplan Meier estimators) for the different Syt1 mutants under normal VAMP2 conditions in the presence of 2 µM Cpx. The corresponding survival curves are shown in **Figure 5A**.

| Time (s) (post-docking) | vSUV | Syt1[WT] | Syt1[F349A] | Syt1[Q] | Syt1[LLQQ] |
|---|---|---|---|---|---|
| 0.588 | 0.6714 | 0.9959 | 0.9936 | 0.9913 | 0.9878 |
| 1.029 | 0.2981 | 0.9936 | 0.9910 | 0.9821 | 0.9810 |
| 2.058 | 0.0617 | 0.9927 | 0.9857 | 0.9734 | 0.9688 |
| 3.087 | 0.0391 | 0.9905 | 0.9794 | 0.9646 | 0.9566 |
| 4.116 | 0.0267 | 0.9852 | 0.9762 | 0.9573 | 0.9471 |
| 5.145 | 0.0243 | 0.9841 | 0.9722 | 0.9445 | 0.9444 |
| 7.497 | 0.0215 | 0.9833 | 0.9597 | 0.9379 | 0.8943 |
| 9.996 | 0.0195 | 0.9824 | 0.9475 | 0.9301 | 0.8808 |

**Table 6.** The survival curves shown in *Figure 5A* were compared pair-wise using the log-rank test to determine the statistical significance (p-values) of the effects of the Syt1 mutants under normal VAMP2 conditions in the presence of 2 μM Cpx.

|  | vSUV | Syt1[WT] | Syt1[F349A] | Syt1[Q] | Syt1[LLQQ] |
|---|---|---|---|---|---|
| v-SUV | n/a | p<0.0001 | p<0.0001 | p<0.0001 | p<0.0001 |
| Syt1[WT] | p<0.0001 | n/a | p=0.971 | p=0.912 | p=0.151 |
| Syt1[F349A] | p<0.0001 | p=0.971 | n/a | p=0.865 | p=0.906 |
| Syt1[Q] | p<0.0001 | p=0.912 | p=0.865 | n/a | p=0.426 |
| Syt1[LLQQ] | p<0.0001 | p=0.151 | p=0.906 | p=0.426 | n/a |

structure to enable ultra-fast fusion (*Li et al., 2019*; *Rothman et al., 2017*; *Volynski and Krishnakumar, 2018*).

We have articulated the simplest hypothesis, considering discrete 'central' and ''peripheral' SNAREpins associated with Syt1 and Cpx, respectively. However, it is easy to imagine that Cpx also binds the central SNAREpins. Indeed, recent X-ray crystal structure revealed that both Syt1 and Cpx can bind the same pre-fusion SNARE complex (*Zhou et al., 2017*). Cpx binding is also predicted to create a new Syt1 binding site (i.e. tripartite interface) on SNAREs. In our reconstituted setup, we find that this SNARE-Cpx-Syt1 'tripartite' interface is not required to produce the fusion clamp but is likely involved in Ca$^{2+}$ activation of fusion from the clamped state (*Figure 5*).

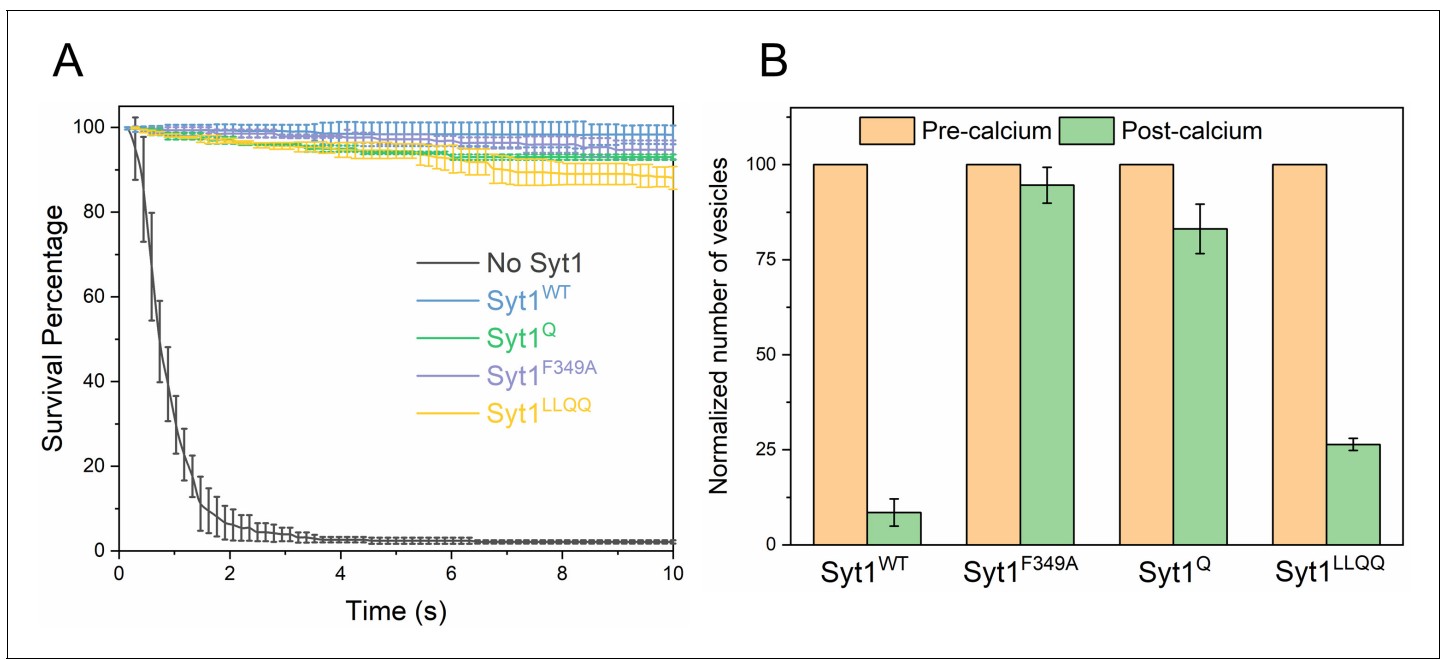

**Figure 5.** Syt1-clamped 'central' SNAREpins are required for Ca$^{2+}$-evoked fusion. The effect of targeted Syt1 mutations was assessed under physiologically-relevant SUV conditions (~74 copies of VAMP2 and ~25 copies of Syt1 wild type or mutant) in the presence of 2 μM Cpx. (**A**) Survival analysis shows neither the disruption of the Syt1-SNARE primary interface (Syt1$^Q$, green curve) and the Syt1-Cpx-SNARE tripartite interface (Syt1$^{LLQQ}$, yellow curve) nor destabilizing Syt1 oligomerization (Syt1$^{F349A}$, purple curve) has any effect on the fusion clamp in the presence of Cpx, with stably docked vesicles observed under all conditions. (**B**) End-point analysis indicates that the Syt1$^Q$ and Syt1$^{F349A}$ vesicles are insensitive to Ca$^{2+}$ and do not fuse upon addition of 1 mM Ca$^{2+}$. In contrast, rapid and synchronous Ca$^{2+}$-triggered fusion was observed with the majority of the docked Syt1$^{WT}$ and Syt1$^{LLQQ}$ vesicles. Notably, the LLQQ mutation does have a small but significant effect on Ca$^{2+}$-triggered release. This implies that the Syt1-Cpx-SNARE tripartite interface is not necessary for clamp, but might be involved in the Ca$^{2+}$-activation process. Overall, the data suggest that the Syt1 clamped SNAREpins are required for Ca$^{2+}$-triggered exocytosis and in the absence of Syt1 clamp, Cpx irreversibly blocks vesicle fusion. The average values and standard deviations from three independent experiments are shown for each condition. In total,~200 vesicles were analyzed for each condition.
The online version of this article includes the following figure supplement(s) for figure 5:

**Figure supplement 1.** Syt1 non-clamping mutants introduce delay in fusion kinetics.

**Figure supplement 2.** Cpx irreversibly blocks SNARE mediated fusion in the absence of Syt1 clamp.

Since the SNARE-Syt1-Cpx tripartite interface is created only when the SNAREs are partially-assembled (i.e. when Cpx binds), it is possible that this interaction plays an auxiliary role in establishing the fusion clamp. As such, it can be bypassed in our biochemically-defined experimental setup. The tripartite interface might become more relevant in the pre-synaptic terminals (*Zhou et al., 2017*) where ~ 30% of Syt1 is present in the PM (*Wienisch and Klingauf, 2006*) and other Synaptotagmins could also participate in the tripartite interface (*Rothman et al., 2017*; *Volynski and Krishnakumar, 2018*; *Zhou et al., 2017*).

Indeed, we have postulated that such a 'dual-clamp' arrangement with Syt1 (from SV) occupying the 'primary' site and with Syt1 and Syt7 (from PM) competing for the 'tripartite' site (in conjunction with Cpx) could potentially explain the synergistic regulation of neurotransmitter release by different Syt isoforms (*Rothman et al., 2017*; *Volynski and Krishnakumar, 2018*). Such an arrangement could also potentially explain how Cpx could regulate $Ca^{2+}$-evoked synchronous neurotransmitter release.

Physiologically, all three synaptic SNAREs (Syntaxin, SNAP-25 and VAMP2) are assembled into a SNAREpin in a concerted reaction involving chaperones Munc13 and Munc18 (*Baker and Hughson, 2016*; *Rizo, 2018*), the need for which is by-passed in our system by pre-assembled t-SNARE complexes. We imagine that these chaperones also function in a concerted manner with Syt1 and Cpx and have recently advanced a speculative model outlining how these proteins can co-operate to template SNAREpin assembly (*Rothman et al., 2017*). Moreover, lipid membranes may contribute to synergism or cooperativity between Syt1 and Cpx in both clamping the un-initiated fusion events and triggering rapid and synchronous fusion in response to $Ca^{2+}$-influx.

As such, further studies with high temporal resolution involving detailed mutational and topological analysis of Syt1 and Cpx, along with the chaperones Munc13 and Munc18, is needed to establish pointillistic details of fast $Ca^{2+}$-triggered SV fusion. Nonetheless, our data demonstrate that Syt1 and Cpx, along with SNARE proteins, form the minimal protein machinery that is necessary and sufficient to establish rapid $Ca^{2+}$-regulated exocytosis.

## Materials and methods

### Materials

The following cDNA constructs, which have been previously described (*Krishnakumar et al., 2013*; *Krishnakumar et al., 2011*; *Weber et al., 1998*; *Mahal et al., 2002*), were used in this study: full-length VAMP2 (VAMP2-His[6], residues 1–116); full-length VAMP2[4X] (VAMP2-His[6], residues 1–116 with L70D, A74R, A81D, L84D mutations), full-length t-SNARE complex (mouse His[6]-SNAP25B, residues 1–206 and rat Syntaxin1A, residues 1–288); soluble cytoplasmic domain of the t-SNAREs (CDT, mouse His[6]-SNAP25B, residues 1–206 and rat Syntaxin1A, residues 1–265); Synaptotagmin (rat Synaptotagmin1-His[6], residues 57–421); and Complexin (human His[6]-Complexin 1, residues 1–134). All mutants including Syt1[F349A] (F349A); Syt1[Q] (R281A/E295A/Y338W/R398A/R399A); Syt1[LLQQ] (L387Q/L394Q); Syt1[DA] (D309A, D363A, D365A) and SNARE[Q] (SNAP25 K40A/D51A/E52A/E55A/D166A) were generated in the above described Syt1 and t-SNARE background respectively using the Quick-Change mutagenesis kit (Agilent Technologies, Santa Clara, CA). Lipids, 1,2-dioleoyl -snglycero-3-phosphocholine (DOPC), 1,2-dioleoyl-sn-glycero-3- (phospho-L-serine) (DOPS), 1,2-dipalmitoyl-sn-glycero-3-phosphoethanolamine-N-(7-nitro-2–1,3-benzoxadiazol-4-yl) (NBD-DOPE), phosphatidylinositol 4, 5-bisphosphate (PIP2) were purchased from Avanti Polar Lipids (Alabaster, AL). ATTO647N-DOPE was purchased from ATTO-TEC, GmbH (Siegen, Germany) and lipophilic carbocyanine DiD (1,1'-Dioctadecyl-3,3,3',3'-Tetramethylindodicarbocyanine Perchlorate) was purchased from Thermofisher Scientific (Waltham, MA). Calcium Green conjugated to a lipophilic 24-carbon alkyl chain (Calcium Green C24) was custom synthesized by Marker Gene Technologies (Eugene, OR).

### Protein expression and purification

All proteins (v- and t-SNAREs, Cpx, Syt1 wild type and mutants) were expressed and purified as described previously (*Mahal et al., 2002*; *Krishnakumar et al., 2013*; *Krishnakumar et al., 2011*; *Weber et al., 1998*). In brief, proteins were expressed in *E. coli* strain Rosetta2(DE3) (Novagen, Madison, WI) using 0.5 mM IPTG for 4 hr. Cells were pelleted and lysed using a cell disruptor (Avestin, Ottawa, Canada) in HEPES buffer (25 mM HEPES, 400 mM KCl, 4% Triton X-100, 10% glycerol, pH 7.4) containing 0.2 mM Tris(2-carboxyethyl) phosphinehydrochloride (TCEP), and 1 mM

phenylmethylsulfonyl fluoride (PMSF). Samples were clarified using a 45Ti rotor (Beckman Coulter, Atlanta, GA) at 40 K RPM for 30 min and subsequently incubated with Ni-NTA resin (Thermofisher Scientific, Waltham, MA) overnight at 4°C. The resin was washed with HEPES buffer (for Cpx) or HEPES buffer supplemented with 1% octylglucoside (Syt1 and SNAREs). Protein was eluted using 350 mM Imidazole and the concentration was determined using a Bradford Assay (BioRad, Hercules, CA) with BSA as a standard. Syt1 was further treated with Benzonase (Millipore Sigma, Burlington, MA) at room temperature for 1 hr, followed by ion exchange (Mono S, AKTA purifier, GE) to remove DNA/RNA contamination. The purity was verified using SDS-PAGE analysis and all proteins were flash frozen and stored at −80°C with 10% glycerol without significant loss of function.

## Liposome preparation

t-SNAREs and VAMP2 (±Syt1) containing SUV were prepared using rapid detergent (1% Octylglucoside) dilution and dialysis method as described previously (*Weber et al., 1998*; *Ji et al., 2010*). VAMP2 (±Syt1) containing SUVs were subjected to additional purification on the discontinuous Nycodenz gradient. The lipid composition was 80 (mole)% DOPC, 15% DOPS, 3% PIP2% and 2% NBD-PE for t-SNARE SUV and 88% DOPC, 10% PS and 2% ATTO647-PE for VAMP2 (±Syt1) SUVs. To mimic physiological copy numbers of protein, we used an input of protein: lipid ratio as 1: 400 for t-SNARE, 1:100 for VAMP2 for physiological density, 1: 500 for VAMP2 at low copy number, and 1: 250 for Syt1. This was based on well-established parameters namely that the reconstitution efficiency for SNAREs and Syt1 is roughly 40–50% (densitometry analysis of the proteoliposomes) and only approximately 50–60% of the proteins are externally oriented (chymotrypsin protection analysis) (*Ji et al., 2010*; *Ramakrishnan et al., 2019*; *Weber et al., 1998*). Based on the densitometry analysis of Coomassie-stained SDS gels, we estimated vesicles at physiological density, contained 74 ± 4 and 26 ± 6 copies of outward-facing VAMP2 and Syt1 respectively (*Figure 1—figure supplement 1*) and vesicles at low copy number of VAMP2 contained 13 ± 2 and 26 ± 6 copies of outward-facing VAMP2 and Syt1 respectively.

## Single vesicle fusion assay

All the single-vesicle fusion measurements were carried out with suspended lipid bilayers as previously described (*Ramakrishnan et al., 2019*; *Ramakrishnan et al., 2018*). Briefly, t-SNARE-containing giant unilamellar vesicles as prepared using the osmotic shock protocol (*Motta et al., 2015*) were busted on freshly plasma-cleaned Si/SiO2 chips containing 5 μm diameter holes in presence of HEPES buffer (25 mM HEPES, 140 mM KCl, 1 mM DTT) supplemented with 5 mM MgCl$_2$. The bilayers were extensively washed with HEPES buffer containing 1 mM MgCl$_2$ and the fluidity of the t-SNARE containing bilayers was verified using fluorescence recovery after photo-bleaching using the NBD fluorescence. In some experiments, we labeled the t-SNAREs with Alexa-488 and confirmed protein mobility as described previously (*Ramakrishnan et al., 2018*).

Vesicles (100 nM lipids) were added from the top using a pipette and allowed to interact with the bilayer for 10 min. We used the ATTO647-PE fluorescence to track vesicle docking, post-docking diffusion, docking-to-fusion delays and spontaneous fusion events. Fusion was attested by a burst and then rapid decrease in fluorescence intensity as the fluorescent PE from the vesicle diffuses away. The time between docking and fusion corresponded to the fusion clamp and was quantified using a 'survival curve' whereby delays are pooled together, and their distribution is plotted in the form of a survival function (*Figure 1*). After the initial 10 min interaction phase, the excess vesicles in the chamber were removed by buffer exchange (3x buffer wash) and 1 mM CaCl$_2$ was added from the top to monitor the effect of Ca$^{2+}$ on the docked vesicles. The number of fused (and the remaining unfused) vesicles was estimated (end-point analysis)~1 min after Ca$^{2+}$-addition.

All experiments were carried out at 37°C using an inverted laser scanning confocal microscope (Leica-SP5) equipped with a multi-wavelength argon laser including 488, diode lasers (532 and 641 nm), and a long-working distance 40X water immersion objective (NA 1.1). The emission light was spectrally separated and collected by photomultiplier tubes. To cover large areas of the planar bilayer and simultaneously record large ensembles of vesicles, the movies were acquired at a speed of 150 ms per frame. Accurate quantification and fate of each vesicles were analyzed using our custom written MATLAB script described previously (*Ramakrishnan et al., 2018*). The files can be downloaded from MATLAB Central at the following website: https://www.mathworks.com/

matlabcentral/fileexchange/66521-fusion-analyzer-fas. Note: We excluded vesicles bound to the edge of the holes as they may not be representative of vesicles bound to the free-floating membrane. We thus used only the centrally-docked vesicles for analysis. We did not observe any change in ATTO-647-PE fluorescence for the vesicles that remain docked and un-fused during the observation period or post $Ca^{2+}$-addition. Thus, we can rule out hemi-fusion diaphragm formation as a possible explanation for the observed 'clamped' or 'un-fused' state.

### Single-vesicle docking analysis

To get an accurate count of the docked vesicles, we used VAMP2 protein with mutations in the C-terminal half (L70D, A74R, A81D and L84D; VAMP2$^{4X}$) that eliminates fusion without impeding the docking process (*Krishnakumar et al., 2013*). For the docking analysis, VAMP2$^{4X}$ containing SUVs (vSUV$^{4X}$) were introduced into the chamber and allowed to interact with the t-SNARE bilayer. After a 10 min incubation, the bilayer was thoroughly washed with running buffer (3x minimum) and the number of docked vesicles were counted. For an unbiased particle count, we employed a custom-written algorithm to count particles from top-left to bottom-right that ensures every spot is counted only once.

### Calcium dynamics

To quantify the kinetics of $Ca^{2+}$-triggered vesicle fusion, we used a $Ca^{2+}$-sensor dye, Calcium Green conjugated to a lipophilic 24-carbon alkyl chain (Calcium Green C24) introduced in the suspended bilayer to directly monitor the arrival of $Ca^{2+}$. Calcium green is a high-affinity $Ca^{2+}$-sensor ($K_d$ of ~75 nM) and exhibits a large fluorescent increase (at 532 nm) upon binding $Ca^{2+}$. To accurately estimate the arrival of $Ca^{2+}$ to/near the vesicles docked on the bilayer, we used confocal microscopy equipped with resonant scanner focused at or near the bilayer membrane and acquired movies at a speed of up to 36 msec per frame. We typically observed the fluorescence signal increase at the bilayer surface between about three frames (~100 msec) after $Ca^{2+}$ addition (*Figure 2—figure supplement 1*). We therefore used 100 msec as the benchmark to accurately estimate the time-constants for the $Ca^{2+}$-triggered fusion reaction.

## Acknowledgements

We thank Dr. Frederic Pincet for critical inputs during the design and development of this project. This work was supported by National Institute of Health (NIH) grant DK027044 to JER.

## Additional information

### Funding

| Funder | Grant reference number | Author |
| --- | --- | --- |
| National Institute of Diabetes and Digestive and Kidney Diseases | 027044 | James E Rothman |

The funders had no role in study design, data collection and interpretation, or the decision to submit the work for publication.

### Author contributions

Sathish Ramakrishnan, Conceptualization, Data curation, Software, Formal analysis, Investigation, Methodology, Writing - review and editing; Manindra Bera, Data curation, Formal analysis, Investigation, Methodology, Writing - review and editing; Jeff Coleman, Resources, Investigation, Methodology, Writing - review and editing; James E Rothman, Conceptualization, Supervision, Funding acquisition, Methodology, Writing - original draft, Project administration, Writing - review and editing; Shyam S Krishnakumar, Conceptualization, Supervision, Funding acquisition, Writing - original draft, Project administration, Writing - review and editing

## Author ORCIDs

Sathish Ramakrishnan  http://orcid.org/0000-0002-7844-2234
Manindra Bera  http://orcid.org/0000-0001-9297-8126
James E Rothman  https://orcid.org/0000-0001-8653-8650
Shyam S Krishnakumar  https://orcid.org/0000-0001-6148-3251

## Decision letter and Author response

Decision letter https://doi.org/10.7554/eLife.54506.sa1
Author response https://doi.org/10.7554/eLife.54506.sa2

# Additional files

## Supplementary files

• Source data 1. Ramakrishan et al_2020_Source Data.

• Transparent reporting form

## Data availability

All data generated or analysed during this study are included in the manuscript and supporting files, including Source data 1.

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
