## [Decision Letter]

**Acceptance summary:**

One of the most compelling new results reported in this work shows that the combination of complexin and synaptotagmin clamps much more than the simple sum of both their clamping functions. This observation may suggest that the two molecules (complexin and synaptotagmin) act cooperatively in their clamping function.

**Decision letter after peer review:**

[Editors’ note: the authors submitted for reconsideration following the decision after peer review. What follows is the decision letter after the first round of review.]

Thank you for submitting your work entitled "Independent Yet Synergistic Roles of Synaptotagmin-1 and Complexin in Calcium Regulated Neuronal Exocytosis" for consideration by *eLife*. Your article has been reviewed by a Senior Editor, a Reviewing Editor, and three reviewers. The reviewers have opted to remain anonymous.

Our decision has been reached after consultation between the reviewers. Based on these discussions and the individual reviews below, we regret to inform you that current manuscript is not acceptable, but we encourage you to consider a resubmission after addressing the concerns raised by the reviewers.

While the reviewers and editors found the new reconstitution method and the data presented very interesting, there appears to be a large gap between the experiments and the model/interpretation presented. Further experiments and a substantial revision of the paper could improve the work and increase its impact.

One of the most compelling new results suggests that combination of complexin and synaptotagmin clamps much more than the simple sum of both their clamping functions. This observation may suggest that the two molecules (complexin and synaptotagmin) act cooperatively in their clamping function. For example, the tripartite interface observed in the crystal structures by Zhou et al., 2017, may produce this cooperativity, but there could be other explanations. Additional mutations should be tested to correlate the various interactions (observed in structures and by the author's previous experiments) with functional studies with their reconstituted assay. We note that while the DA and Q mutants have been tested extensively in different contexts (these mutants primarily interfere with calcium binding and formation of the primary interface, respectively) the F349A mutation is less clearly defined since it may also affect (positively or negatively) the tripartite interface (it is an integral part of largely hydrophobic interactions between Syt1 C2B and the SNARE complex through the tripartite interaction, please consult the crystal structure of this complex). Mutations should be designed that more precisely test the oligomeric Syt1 interactions and the tripartite interface.

*Reviewer #1:*The authors have used a pore-spanning lipid bilayer setup to study the functional interplay between small unilamellar vesicles (SUVs) bearing both the v-SNARE VAMP2 and synaptotagmin (Syt1) and target membranes bearing pre-formed t-SNAREs, with or without complexin (Cpx). Previously (Ramakrishnan, 2019), the authors showed that the presence of Syt1 on the VAMP2-bearing SUVs led to 'clamped' vesicles – that is, vesicles that docked and remained immobile, without fusing, for at least ten minutes. In Figure 4 of this manuscript, they reproduce the experiments shown in Figure 3 of the earlier manuscript, albeit with two additional Syt1 mutants. Most of the current manuscript, however, is founded on the new observation that using a five-fold higher – and more physiologically realistic – number of VAMP2 per SUV dramatically alters the results. Instead of Syt1 being sufficient for clamping, its presence now merely delays fusion modestly. Strikingly the addition of Cpx, which in the absence of Syt1 has no effect, leads to an increase in the number of docked vesicles, and >90% of them (as opposed to 20% without Cpx) are securely clamped. The clamp so formed is efficiently reversed by Ca^2+^.

This leads to a model in which some SNARE complexes are clamped by Syt1 and others are clamped by Cpx. Since, in these experiments, Syt1 is outnumbered by VAMP2 by about 3:1, it stands to reason that it can't do all of the needed clamping on its own. (In Ramakrishnan, 2019, there was twice as much Syt1 as VAMP2). When the authors washed out Cpx the vesicles, presumably using the excess SNAREs, fused; when the Cpx was washed out in the presence of a reagent that trapped the liberated SNAREs in an inactive state, fusion was not observed until the Syt1 clamp was released by Ca^2+^ addition.

The authors' results imply that Syt1 is somehow required for Cpx to function as a fusion clamp, since Cpx alone shows no discernable clamping activity. Their model to explain this is that Syt1 forms a ring-like oligomer that impedes full zippering of the other 'peripheral' SNARE complexes, giving Cpx the opportunity to bind and clamp them. The problem with this model is that the only experiment that might be considered to be a critical test – using the F349A mutant to destabilize the Syt1 oligomers – not only doesn't abrogate Cpx-mediated clamping, but renders Cpx clamping irreversible. I don't know how to make sense of this. More generally, since Syt1:Cpx 'synergy' is quite central to the manuscript, one could argue that the authors need to investigate further; for example, by testing Syt1 mutations that disrupt the tripartite complex.*Reviewer #2:*Ramakrishnan and colleagues follow up on a study of theirs published earlier this year (FEBS Letters 2019) examining synaptic vesicle (SV) clamping mechanisms using a minimal reconstituted liposome fusion system comprising SNAREs, Syt1, and Cpx1. The current results indicate separate clamping functions for Syt1 and Cpx1, and both are required to produce calcium-sensitive stably docked vesicles in the context of physiological VAMP2 and Syt1 copy numbers. Disruption of a binding interface between the tSNAREs and C2B prevents Syt1-mediated clamping but also reveals an apparently irreversible clamping function of Cpx1. I am generally enthusiastic about the work, but I do have some questions on data interpretation and the proposed model.

From a technical perspective, this is a nice study that further develops a powerful in vitro docking/fusion model sufficient for detailed exploration of Syt1/Cpx1/SNARE mechanisms. Several of the interpretations put forward by the authors could use some clarification. In particular, the conclusion that Cpx1 acts independently of Syt1 on a distinct set of peripheral SNAREpins to clamp fusion is emphasized throughout the manuscript but not strongly supported by the data. On its own, Cpx1 does not provide a stable (ie 10's of seconds to minutes time scale) clamping function in this assay. But mutations in the 'primary' C2B-tSNARE interface eliminate Syt1 clamping function while somehow permitting an unexpected and potent Cpx1 stable clamping function. Also, previous work has suggested two distinct binding interfaces between C2B and the tSNAREs: the primary and tripartite interfaces (Zhou et al., 2017). The tripartite interface involves Cpx1 and may provide an explanation for the stable clamping by Cpx1 when the primary C2B interface is disrupted. Since C2B domains may interact with SNAREpins in these two distinct interfaces, changing the VAMP2:Syt1 ratio may alter the relative fraction of primary vs tripartite binding interactions and provide another explanation for the dose-dependence of VAMP2 on sufficiency of Syt1 for clamping versus the requirement for Cpx1 in the face of excess VAMP2. These scenarios may not be consistent with the picture of a central set of SNAREpins harboring Syt1 surrounded by Cpx1-bound SNAREpins as promoted by the authors. And no direct evidence for these central/peripheral SNAREpin arrangements is provided here. Thus, such a strong emphasis on independence of Syt1 and Cpx1 mechanisms in the Abstract, Results section and Discussion section is not warranted in my opinion.

In the high VAMP2:Syt1 experiments, the authors elegantly demonstrated the reversible nature of the Cpx1 clamping function by washing out Cpx1 and competing away excess SNAREpins with soluble tSNARE complexes. Did the authors attempt to wash out Cpx1 in the Syt1(Q)+tSNARE(Q) experiments where Cpx1 appeared to drive an irreversible clamp? And given the possible contribution of the tripartite C2B-SNARE interaction, did the authors attempt to disrupt this interface with mutations published in the Zhou et al., 2017 study? I do not think additional experiments would be required for publication of the current study, but could help bolster the proposed model.

The authors emphasize the importance of physiological copy number for the VAMP2 and Syt1 used in this study. How important is it to have 25 copies of (outwardly facing) Syt1 per vesicle? Work from Jahn put the number at around 15 copies per glutamatergic SV and possibily even fewer for GABAergic SVs. Does the clamping efficiency gradually go down as the copy numbers are reduced or is there a minimal requirement for any clamping to be observed? Perhaps other sources of C2 domains could contribute in synapses, but I was anticipating some comment on this 25 copy number given the Jahn work.

*Reviewer #3:*This manuscript explores the functional interactions between the Ca^2+^ sensor, Synaptotagmin1 (Syt1) and Complexin (Cpx) in clamping of neuronal SNARE complexes preceding Ca^2+^ influx and rapid fusion. They use a version of an in vitro single molecule fusion assay, which uses v-SNARE containing SUVs, with Si/SiO2 chips with t-SNARE containing membranes spread across the holes. Here, the authors examine membrane fusion between VAMP2-SUVs w/ and w/o Syt1, +/- Cpx, with t-SNARE complex membranes. Using a series of different protein combinations and concentrations, and various Syt1 mutant proteins, the authors suggest that two types of clamped SUVs occur: a small central "core" that is clamped by Syt1, and the rest that are clamped by Cpx, but only when facilitated by Syt1. Furthermore, they suggest that addition of Ca^2+^ facilitates fusion of only the Syt1-clamped SUVs, and that Cpx-clamped SUVs are blocked from fusion. It is an interesting hypothesis, but not well supported by the data presented.

A substantial amount of the data in the paper is basically validating their assay under "physiological conditions" of ~70 VAMPs and ~25 Syt1s per SUV, +/- Cpx (2uM), with the number of t-SNARE complexes unknown. The authors build on previous data from their lab, as well as others, related to the idea that Syt1 and Cpx work both independently as well as synergistically. The logic of the conclusions is based on a number of assumptions based on previous work, especially with regard to oligomerization of Syt1, interactions with Cpx and SNAREs, and whether washing 40x only serves to displace Cpx. This is particularly an issue in assays that are quite sensitive to the amount of protein used, but the extent of complex formation is only inferred indirectly. The authors present a hypothesis to try and explain the complicated findings, yet it is unclear whether the data fully support such a hypothesis. One particular example is the finding in Figure 1 that SUVs with Cpx alone fuse relatively quickly, yet the conclusion from Figure 4 is that "in the absence of the Syt1 clamp, Cpx blocks SNARE assembly irreversibly".

For most experiments, only a representative curve is shown, or the data (late time point?) summarized in bar graphs, with little detail of kinetics or binding shown. In many cases, data are "normalized," but it is unclear what the data are normalized to. It is also unclear if contents mixing experiments are performed for each of the different assay conditions, or only a representative experiment was done. These issues make many of the experiments tricky to interpret, and it is quite possible that alternative hypotheses could explain the data.

Furthermore, there is a lack of sufficient discussion (and referencing) of a number of other studies in the field. The authors need to try to reconcile their data with that of other groups, in order for their new hypothesis of Syt1-Cpx to be at all convincing.

[Editors’ note: further revisions were suggested prior to acceptance, as described below.]

Thank you for submitting your article "Independent Yet Synergistic Roles of Synaptotagmin-1 and Complexin in Calcium Regulated Neuronal Exocytosis" for consideration by *eLife*. Your article has been reviewed by Vivek Malhotra as the Senior Editor, a Reviewing Editor (Axel Brunger), and three reviewers. The reviewers have opted to remain anonymous.

The reviewers have discussed the reviews with one another and the Reviewing Editor has drafted this decision to help you prepare a revised submission. Please aim to submit the revised version within two months, but we are happy to extend this timeframe if needed. We recognize that we live in difficult and unprecedented times, so we are prepared to hear from you about a longer than usual time period for revisions.

Summary:

We thank the authors for responding to our previous concerns and including additional experiments. The hypothesis that there are distinct roles for Cpx and Syt1 in clamping two populations of SNAREs on a single vesicle is interesting and novel. However, this revised manuscript will benefit from substantial further revisions, clarifications, and discussion of other models and explanations. Moreover, rigorous statistical analyses of the survival times are requested. Our chief concerns are centered on the proposed model, interpretations, and analysis of the data, and no new experiments are requested. Below we detail several revisions required by the reviewers.

Essential revisions:

1) We are not convinced that the interpretation of their data is consistent with regard to the 'independent' roles of Cpx and Syt1 as stated in the title. Cpx on its own cannot clamp vesicles whereas in the presence of even a defective version of Syt1 (either the oligomer mutation or the quintuple Syt1+quintuple SNAP-25 mutation), Cpx now permanently inhibits vesicles from fusing. Thus, Cpx still clearly depends on Syt1 in a way that is not consistent with either the oligomer assembly or the primary Syt1-SNARE interface. One is left with the puzzle of how precisely synaptotagmin is creating this delay. Whatever this kinetic effect is, Cpx is not functioning independently of Syt1. Please revise the Results section and Discussion section to clarify your hypothesis and to better define independence in this context.

2) The notion of "clamping" suggests that an underlying molecular mechanism involving the synaptic proteins that have been included in this study. Yet, the measurements are assessing the survival of docked vesicles which is not a direct measure of molecular clamping. While molecular clamping could indeed be a possible explanation for the observations, there could be other explanations as well that involve the interplay between molecules and membranes. For example, it is possible that synaptotagmin or SNARE-induced hemifusion diaphragm formation (a long-lived metastable state) could affect the time of survival. Another possibility is that the membrane itself is the conduit for synergism. For example, it is known that synaptotagmin bends membranes and complexin preferentially binds to curved membranes, and thus, the membrane may introduce apparent synergism or cooperativity between the proteins. Please revise the Abstract, Results section and Discussion section accordingly.

3) The discussion in the text leaves one with the impression that the Syt1 C2B LLQQ mutant has no effect in their assay. Actually, it does have a significant effect, see Figure 4—figure supplement 1 panel B – the effect is somewhere between the quintuple mutant and wildtype after calcium addition. Please discuss.

4) The "survival percentage" plots are not quantitative. Please provide bar charts with error bars (along with significance tests) of the survival percentages after some defined time period(s). Another suggestion is to provide survival statistics (such as Kaplan Meier estimators).

5) Please promote some of the supplemental figures (especially the supplements to Figure 4) to primary figures, to in order to avoid 'burying' some of the most important results.

[Editors' note: further revisions were suggested prior to acceptance, as described below.]

Thank you for resubmitting your work entitled "Synergistic Roles of Synaptotagmin-1 and Complexin in Calcium Regulated Neuronal Exocytosis" for further consideration by *eLife*. Your revised article has been evaluated by Vivek Malhotra (Senior Editor) and a Reviewing Editor.

We thank the authors for addressing the concerns of the reviewers and reviewing editor. The manuscript has much improved. However, there is a remaining point on statistical analysis that needs to be addressed before final acceptance, as outlined below:

The authors still do not provide some statistical statement of significance in the data presented in Figure 3, Figure 4, and Figure 5. They do provide info on the Kaplan Meier estimator of survival probability, but the reader is left to assume that all the differences shown are significant. They are very likely to be significant because the effects are quite large, but one usually expects a statistical test with either a p value or confidence interval to bolster the claim that an experimental manipulation did or did not have an effect.

---

## [Author Response]

[Editors’ note: The authors appealed the original decision. What follows is the authors’ response to the first round of review.]

While the reviewers and editors found the new reconstitution method and the data presented very interesting, there appears to be a large gap between the experiments and the model/interpretation presented. Further experiments and a substantial revision of the paper could improve the work and increase its impact.One of the most compelling new results suggests that combination of complexin and synaptotagmin clamps much more than the simple sum of both their clamping functions. […] Mutations should be designed that more precisely test the oligomeric Syt1 interactions and the tripartite interface.

To assess if the SNARE-Cpx-Syt1 ‘tripartite’ interaction^1^ could explain the observed cooperative function of Syt1 and Cpx, we tested the effect of the previously described Syt1 mutations (L387Q/L394Q) that disrupt this tripartite interface^1^ in our reconstituted assay. The LLQQ mutation had very little to no effect on the Syt1 clamp or the Ca^2+^-triggered fusion under both the low-copy VAMP2 (without Cpx) and the normal VAMP2 (with 2 µM Cpx) conditions. This suggests that the tripartite interface likely plays an auxiliary role in establishing the Syt1 clamp and cannot explain the observed synergistic effect of Syt1 and Cpx in clamping SNARE-mediated fusion.

These data are now included in Figure 4 and Figure 4—figure supplement 1 of the revised manuscript. Additionally, we have now expanded the Discussion section as follows: “We find that the SNARE-Cpx-Syt1 ‘tripartite’ interface, which has been shown to be physiologically-relevant ^1^, is not absolutely required to produce the Syt1 clamp or for the Ca^2+^ activation process under reconstituted conditions (Figure 4). […] Nonetheless, our data demonstrates that Syt1 and Cpx, along with SNARE proteins, form the minimal protein machinery that is necessary and sufficient to establish rapid Ca^2+^-regulated exocytosis”.

With regards to the F349A mutation, we have previously shown that this mutation specifically disrupts Syt1 oligomerization without affecting other molecular properties, including overall SNARE binding^7^. However, its effect on the ‘primary’ or the ‘tripartite’ SNARE binding sites has not been resolved. Nonetheless, considering that the LLQQ mutation has no effect on Syt1 clamping or activation function, it is unlikely that the F349A mutation exerts its effect via the tripartite interface. We thus conclude that Syt1 oligomerization is a key element of the Syt1 clamp.

Reviewer #1:The authors' results imply that Syt1 is somehow required for Cpx to function as a fusion clamp, since Cpx alone shows no discernable clamping activity. Their model to explain this is that Syt1 forms a ring-like oligomer that impedes full zippering of the other 'peripheral' SNARE complexes, giving Cpx the opportunity to bind and clamp them. The problem with this model is that the only experiment that might be considered to be a critical test – using the F349A mutant to destabilize the Syt1 oligomers – not only doesn't abrogate Cpx-mediated clamping, but renders Cpx clamping irreversible I don't know how to make sense of this.

Survival analysis shows that the Syt1^Q^ (SNARE ‘primary’ interface) and Syt1^F349A^ (oligomerization) mutants destabilize the Syt1 clamp and abrogate the formation of central SNAREpins but still introduce a meaningful delay in the overall fusion process. This kinetic delay is sufficient for Cpx to arrest the assembly of all available SNAREs. Thus, we observe an irreversible clamped state under these conditions. This implies that the kinetic delay introduced by Syt1, irrespective of the Syt1 clamp, is adequate for Cpx clamping function.

These data are now included in Figure 4—figure supplement 2 and we have revised the relevant Results section as follows: “We also tested the effect of the Syt1 mutants using vesicles containing physiological VAMP2 and Syt1 copy numbers in the presence of 2 µM Cpx. […] This denotes that in Syt1^Q^ vesicles (and presumably in Syt1^F349A^ vesicles), all SNAREpins are clamped by Cpx alone and become accessible to CDT block following the Cpx wash-out”

More generally, since Syt1:Cpx 'synergy' is quite central to the manuscript, one could argue that the authors need to investigate further; for example, by testing Syt1 mutations that disrupt the tripartite complex.

We have now tested and confirmed that the tripartite interaction cannot explain the Syt1/Cpx synergy. See question #1 for detailed response.

Reviewer #2:On its own, Cpx1 does not provide a stable (ie 10's of seconds to minutes time scale) clamping function in this assay. But mutations in the 'primary' C2B-tSNARE interface eliminate Syt1 clamping function while somehow permitting an unexpected and potent Cpx1 stable clamping function.

We find that Syt1 even in the absence of a stable clamp (i.e. Syt1^Q^ and Syt1^F349A^) introduces a meaningful delay in the SNARE-mediated fusion and this kinetic delay is sufficient/required for Cpx to act as a fusion clamp. See question #2 for detailed response.

The tripartite interface involves Cpx1 and may provide an explanation for the stable clamping by Cpx1 when the primary C2B interface is disrupted. Since C2B domains may interact with SNAREpins in these two distinct interfaces, changing the VAMP2:Syt1 ratio may alter the relative fraction of primary vs tripartite binding interactions and provide another explanation for the dose-dependence of VAMP2 on sufficiency of Syt1 for clamping versus the requirement for Cpx1 in the face of excess VAMP2. Results section, Discussion section.In the high VAMP2:Syt1 experiments, the authors elegantly demonstrated the reversible nature of the Cpx1 clamping function by washing out Cpx1 and competing away excess SNAREpins with soluble tSNARE complexes. And given the possible contribution of the tripartite C2B-SNARE interaction, did the authors attempt to disrupt this interface with mutations published in the Zhou et al., 2017 study?

We have now tested and confirmed that the tripartite interface is not involved in establishing the Syt1 clamp or Ca^2+^-activation mechanisms. Hence, this cannot explain the observed co-operative function of Syt1 and Cpx. See question #1 for detailed response.

These scenarios may not be consistent with the picture of a central set of SNAREpins harboring Syt1 surrounded by Cpx1-bound SNAREpins as promoted by the authors. And no direct evidence for these central/peripheral SNAREpin arrangements is provided here. Thus, such a strong emphasis on independence of Syt1 and Cpx1 mechanisms in the Abstract, Results section, and Discussion section is not warranted in my opinion.

We find that (i) Syt1 and Cpx act synergistically to block spontaneous fusion (ii) Syt1 and Cpx likely act independently and produce molecularly distinct clamped SNAREpins; (iii) the kinetic delay introduced by Syt1, independent of that Syt1 clamp, is need for Cpx to irreversibly arrest SNARE assembly; and (iv) SNARE-Syt1-Cpx tripartite interface cannot explain the observed co-operative function of Syt1 and Cpx (Figure 4). We believe that the outlined central/peripheral SNAREpins associated with Syt1 and Cpx offers the best explanation for all observed data.

Did the authors attempt to wash out Cpx1 in the Syt1(Q)+tSNARE(Q) experiments where Cpx1 appeared to drive an irreversible clamp?

In light of reviewer’s comment, we have carried the Cpx washout experiments under Syt1^Q^ + SNARE^Q^. We find that in the absence of CDT, the Syt1^Q^ vesicle proceed to fuse spontaneously following the buffer wash. Inclusion of CDT (with Cpx washout) irreversibly blocks all fusion events, even in the presence of 1 mM Ca^2+^. This, taken together with data shown in Figure 4, indicates that (i) the Syt1 clamp and the associated central SNAREpins are needed for Ca^2+^-triggered fusion and (ii) Cpx irreversibly blocks SNARE-mediated fusion. These data are included as Figure 4—figure supplement 3 in the revised manuscript and discussed in the Results section.

8) The authors emphasize the importance of physiological copy number for the VAMP2 and Syt1 used in this study. How important is it to have 25 copies of (outwardly facing) Syt1 per vesicle? Work from Jahn put the number at around 15 copies per glutamatergic SV and possibily even fewer for GABAergic SVs. Does the clamping efficiency gradually go down as the copy numbers are reduced or is there a minimal requirement for any clamping to be observed? Perhaps other sources of C2 domains could contribute in synapses, but I was anticipating some comment on this 25 copy number given the Jahn work.

Based on the available literature (Takamori et al., 2014; Wilhelm et al., 2014), there are ~16-22 copies of Syt1 per synaptic vesicle. We chose reconstitution conditions to produce 20-25 copies of outward facing Syt1 as a reasonable approximation. We are currently systematically testing the importance of the Syt1 and VAMP2 copy number. Preliminary analysis indicates that Syt1 clamping and Ca^2+^-triggered fusion is related to the Syt1 copy number and the Syt1:VAMP2 ratio. This will be focus of a future manuscript and as such, we chose not to address this in the current report.

Reviewer #3:[…] A substantial amount of the data in the paper is basically validating their assay under "physiological conditions" of ~70 VAMPs and ~25 Syt1s per SUV, +/- Cpx (2uM), with the number of t-SNARE complexes unknown. The authors build on previous data from their lab, as well as others, related to the idea that Syt1 and Cpx work both independently as well as synergistically. The logic of the conclusions is based on a number of assumptions based on previous work, especially with regard to oligomerization of Syt1, interactions with Cpx and SNAREs, and whether washing 40x only serves to displace Cpx. This is particularly an issue in assays that are quite sensitive to the amount of protein used, but the extent of complex formation is only inferred indirectly.

We made no *a priori* assumptions and systematically tested the effect of Cpx and Syt1 on SNAREmediated fusion. We have examined the role of well-known Syt1 molecular properties, namely ‘primary’ and ‘tripartite’ SNARE binding, oligomerization and Ca^2+^-binding, in relation to the Syt1 clamp and its reversal by Ca^2+^. We advance the ‘synergistic clamping’ model as it best explains all available data (both in vitro and physiology) and is consistent with well-established structural and biochemical properties of Syt1, Cpx and SNAREs. As we have noted, we have put forward the simplest hypothesis to explain our findings and are aware of possible variations/alterations of our model. Nonetheless, we are confident of our central findings that (i) Syt1 and Cpx act independently and produce molecularly-distinct clamped SNAREpins under a single vesicle and (ii) the Syt1-associated ‘central’ SNAREpins are critical and likely sufficient for Ca^2+^-evoked fast vesicular release.

Based on our reconstitution conditions, we estimate that there are ~15,000 freely diffusing t-SNARE complex per 5 µm suspended bilayer. Thus, t-SNARE concentration is not a limiting factor in these experiments. It is worth noting that the t-SNARE concentration on the pre-synaptic membrane is not well-defined and is predicted to be in excess and thus, freely available. As noted, in most experiments, we use physiological copy numbers of Syt1and VAMP2 per vesicles. We chose 2 µM of soluble Cpx in our assay, considering its affinity to pre-fusion SNARE complexes (K_d_ ~ 0.5 µM) and the typical concentration (2-5 µM) used in previous studies (Malsam et al., 2012; Diao et al., 2012; Lai et al., 2014). Hence, we strongly believe our reconstitution conditions are a reasonable approximation of physiological-conditions.

The authors present a hypothesis to try and explain the complicated findings, yet it is unclear whether the data fully support such a hypothesis. One particular example is the finding in Figure 1 that SUVs with Cpx alone fuse relatively quickly, yet the conclusion from Figure 4 is that "in the absence of the Syt1 clamp, Cpx blocks SNARE assembly irreversibly".

Closer inspection of our data reveals that the delay introduced by Syt1, independent of the Syt1 clamp, is required for Cpx to function as a fusion clamp. We have revised our results and discussion accordingly. See question #2 for detailed response.

For most experiments, only a representative curve is shown, or the data (late time point?) summarized in bar graphs, with little detail of kinetics or binding shown. In many cases, data are "normalized," but it is unclear what the data are normalized to. It is also unclear if contents mixing experiments are performed for each of the different assay conditions, or only a representative experiment was done.

All data shown, including the curves, correspond to the average data (± standard deviations) and not representative data. The number of independent trials and total vesicles imaged is included in every figure legend and we will include relevant raw data in the source file. In addition, we have included representative images and video for selected data for clarity and transparency.

We indeed show vesicle fusion kinetics under both basal and Ca^2+^ conditions (Figure 1B and 2C). We do not have single-molecule resolution in our current setup to track protein assembly kinetics. It is worth noting that the protein concentrations used in the reconstituted experiments are comparable to those used by other labs. Considering the binding affinities and the temporal resolution of the experiments, we believe that the protein assembly kinetics should little to no effect on the clamping and fusion properties measured in this study.

We use normalization, with the number of docked vesicles set to 100%, for easy comparison between different assay conditions (Figure 3) and Syt1 mutants (Figure 4). For example, as the number of docked vesicles vary between different mutants, normalization allows us to best illustrate the effect of a given mutation(s) on spontaneous fusion and Ca^2+^-triggered fusion. We will include raw data used in these calculations in the source file.

We carried the content release experiments for the Syt1^WT^ and few chosen mutations to test and qualitatively verify the findings from the lipid mixing analysis. We used lipid-mixing data for all statistical analysis as the fluorescence properties of the Alexa647, which is far superior compared to Sulphorhodamine B, is best suited for our automated analysis. We thus chose to only include a representative content release data for illustrative purposes.

[Editors’ note: what follows is the authors’ response to the second round of review.]

Essential revisions:1) We are not convinced that the interpretation of their data is consistent with regard to the 'independent' roles of Cpx and Syt1 as stated in the title. Cpx on its own cannot clamp vesicles whereas in the presence of even a defective version of Syt1 (either the oligomer mutation or the quintuple Syt1+quintuple SNAP-25 mutation), Cpx now permanently inhibits vesicles from fusing. Thus, Cpx still clearly depends on Syt1 in a way that is not consistent with either the oligomer assembly or the primary Syt1-SNARE interface. One is left with the puzzle of how precisely synaptotagmin is creating this delay. Whatever this kinetic effect is, Cpx is not functioning independently of Syt1. Please revise the Results section and Discussion section to clarify your hypothesis and to better define independence in this context.

The intent was to convey our central finding that Syt1 and Cpx play distinct roles in clamping different pools of SNAREpins. We recognize that the original title might be misleading. As such, we have removed the reference to the ‘independent’ role from the title and revised it to read: Synergistic Roles of Synaptotagmin-1 and Complexin in Calcium Regulated Neuronal Exocytosis. In addition, we have made minor edits to the Abstract and in the Results section Discussion section to clearly explicate our findings.

2) The notion of "clamping" suggests that an underlying molecular mechanism involving the synaptic proteins that have been included in this study. Yet, the measurements are assessing the survival of docked vesicles which is not a direct measure of molecular clamping. While molecular clamping could indeed be a possible explanation for the observations, there could be other explanations as well that involve the interplay between molecules and membranes. For example, it is possible that synaptotagmin or SNARE-induced hemifusion diaphragm formation (a long-lived metastable state) could affect the time of survival. Another possibility is that the membrane itself is the conduit for synergism. For example, it is known that synaptotagmin bends membranes and complexin preferentially binds to curved membranes, and thus, the membrane may introduce apparent synergism or cooperativity between the proteins. Please revise the Abstract, Results section and Discussion section accordingly.

In this study, we define ‘clamping as the survival of docked vesicles beyond the initial observation period. Nonetheless, we find compelling evidence towards the existence of a *‘molecular clamp’*. For example, we observe the fusion clamp *only* in the presence of Syt1 and Cpx and the fusion clamp is altered by disruption of specific molecular properties of Syt1 (i.e. SNARE primary binding, oligomerization). It is quite possible other factors, including membranes, play a role in augmenting the observed synergistic effect of Syt1 and Cpx. This is a very interesting possibility, with wideranging implication and will be focus of our future work. As such, this is beyond the scope of the current work. We have now highlighted this possibility in the Discussion section and included the following statement: Moreover, lipid membranes may contribute to synergism or cooperativity between Syt1 and Cpx in both clamping the un-initiated fusion events and triggering rapid and synchronous fusion in response to Ca^2+^-influx.

Nevertheless, we wish to point out that we do not believe that synaptotagmin or SNARE-induced hemi-fusion diaphragm formation is a likely explanation for the observed clamped state. Hemi-fusion would result in partial-lipid mixing and this can be readily verified by reduction of the vesicle fluorescence (ATTO647-DOPE) and spreading of fluorescence in the outer ROI. We do NOT observe any change in vesicle fluorescence signal for the docked/un-fused vesicles. Thus, we can rule out this possibility. We have explicated this in the revised Materials and methods section as follows: We did not observe any change in ATTO-647-PE fluorescence for the vesicles that remain docked and un-fused during the observation period or post Ca^2+^-addition. Thus, we can rule out hemi-fusion diaphragm formation as a possible explanation for the observed ‘clamped’ or ‘un-fused’ state.

3) The discussion in the text leaves one with the impression that the Syt1 C2B LLQQ mutant has no effect in their assay. Actually, it does have a significant effect, see Figure 4—figure supplement 1 panel B – the effect is somewhere between the quintuple mutant and wildtype after calcium addition. Please discuss.

With regards to the LLQQ mutational analysis, we observe very little to no effect of LLQQ mutation on the fusion clamp under both the low-copy VAMP2 (without Cpx) and the normal VAMP2 (with 2 µM Cpx) conditions. Thus, we conclude that the tripartite site is not essential for the fusion clamp.

The Syt1 LLQQ mutant does have a small, but significant effect in calcium triggering in the presence of Cpx. However, this effect was observed only under physiological VAMP2 conditions (Figure 5), but not under low copy VAMP2 conditions (Data not shown). This *does* suggest that the Syt1-Cpx-SNARE tripartite interaction might be important for calcium triggering mechanism, but it is not conclusive. Indeed, we are investigating this in further detail, including testing the effect of Syt1LLQQ and Syt1 quintuple mutation under varying calcium concentrations and VAMP2/Syt1 ratio with higher temporal resolution.

We do agree that we had not adequately highlighted/discussed the effect of LLQQ mutation on calcium triggering of fusion in the current manuscript. In addition to moving the relevant data into the primary figures (Figure 5), we have revised the Results section as follows: “However, the clamped Syt1Q and Syt1F349A vesicles were insensitive to Ca^2+^ and did not fuse following Ca^2+^ (1 mM) addition as opposed to the rapid and synchronous fusion observed with the majority of the Syt1WT and Syt1LLQQ vesicles (Figure 5). Notably, a significant partial fraction (~25%) of the Syt1LLQQ vesicles remained un-fused even following Ca^2+^ addition (Figure 5). This suggests that while the Syt1- Cpx-SNARE tripartite interface is not essential for establishing the fusion clamp, it is likely important in the Ca^2+^-activation mechanism”.

In addition, we have updated the relevant section in the Discussion section as follows: “We have articulated the simplest hypothesis, considering discrete ‘central’ and ‘’peripheral’ SNAREpin associated with Syt1 and Cpx, respectively. […] The tripartite interface might become more relevant in the pre-synaptic terminals (Zhou et al. 2017) where ~ 30% of Syt1 is present in the plasma membrane (Wienisch and Klingauf 2006) and other Synaptotagmins could also participate in the tripartite interface (Rothman et al. 2017; Volynski and Krishnakumar 2018; Zhou et al. 2017)”.

4) The "survival percentage" plots are not quantitative. Please provide bar charts with error bars (along with significance tests) of the survival percentages after some defined time period(s). Another suggestion is to provide survival statistics (such as Kaplan Meier estimators).

We wish to clarify that all survival curves shown in the manuscript are based on Kaplan Meier estimate calculations and plotted as ‘survival percentage’. We have now made this explicit in the Materials and methods section of the revised manuscript.

As suggested, we have also included the survival statistics (i.e. Kaplan Meier estimators) at defined time-period periods post-docking. These are shown in Table 1, Table 2 and Table 3 in the revised manuscript.

In addition, we have now included expanded (1h) survival plots for vSUVs clamped by Syt1 and Cpx. This is shown in Figure 1—figure supplement 4 of the revised manuscript.

5) Please promote some of the supplemental figures (especially the supplements to Figure 4) to primary figures, to in order to avoid 'burying' some of the most important results.

As recommended, we have promoted Figure 4—figure supplement 1 to the primary Figure 5.

[Editors’ note: what follows is the authors’ response to the second round of review.]

The authors still do not provide some statistical statement of significance in the data presented in Figure 3, Figure 4, and Figure 5. They do provide info on the Kaplan Meier estimator of survival probability, but the reader is left to assume that all the differences shown are significant. They are very likely to be significant because the effects are quite large, but one usually expects a statistical test with either a p value or confidence interval to bolster the claim that an experimental manipulation did or did not have an effect.

We now submit a revised manuscript incorporating the statistical test (log-rank test) of the survival curves shown in Figure 1B, Figure 4A and Figure 5A as recommended by the reviewer.

The statistical significance (p-values) for the pairwise comparison of the survival curves are shown in Table 2, Table 4 and Table 6 of the revised manuscript.